# PROVABLY IMPROVED CONTEXT-BASED OFFLINE META-RL WITH ATTENTION AND CONTRASTIVE LEARNING

## ABSTRACT

Meta-learning for offline reinforcement learning (OMRL) is an understudied problem with tremendous potential impact by enabling RL algorithms in many real-world applications. A popular solution to the problem is to infer task identity as augmented state using a context-based encoder, for which efficient learning of robust task representations remains an open challenge. In this work, we provably improve upon one of the SOTA OMRL algorithms, FOCAL, by incorporating intra-task attention mechanism and inter-task contrastive learning objectives, to robustify task representation learning against sparse reward and distribution shift. Theoretical analysis and experiments are presented to demonstrate the superior performance and robustness of our end-to-end and model-free framework compared to prior algorithms across multiple meta-RL benchmarks. [1]

## 1 INTRODUCTION

Deep reinforcement learning (RL) has achieved many successes with human- or superhuman-level performance across a wide range of complex domains (Mnih et al., 2015; Silver et al., 2017; Vinyals et al., 2019; Ye et al., 2020). However, all these major breakthroughs focus on finding the best-performing strategy by trial-and-error interactions with a single environment, which poses severe constraints for scenarios such as healthcare (Gottesman et al., 2019), autonomous driving (Shalev-Shwartz et al., 2016) and controlled-environment agriculture (An et al., 2021; Cao et al., 2021) where safety is paramount. Moreover, these RL algorithms require tremendous explorations and training samples, and are also prone to over-fitting to the target task (Song et al., 2019; Whiteson et al., 2011), resulting in poor generalization and robustness. To make RL truly practical in many real-world applications, a new paradigm with better safety, sample efficiency and generalization is in need.

Offline meta-RL, as a marriage between offline RL and meta-RL, has emerged as a promising candidate to address the aforementioned challenges. Like supervised learning, offline RL restricts the agent to solely learn from fixed and limited data, circumventing potentially risky explorations. Additionally, offline algorithms are by nature off-policy, which by reusing prior experience, have proven to achieve far better sample efficiency than on-policy counterparts (Haarnoja et al., 2018).

Meta-RL, on the other hand, exploits the shared structure of a distribution of tasks and enables the agent to adapt to new tasks with minimal data. One popular approach is by learning a single universal policy conditioned on a latent task representation, known as context-based method (Hallak et al., 2015). Alternatively, the shared skills can be learned with a meta-controller (Oh et al., 2017).

In this work we restrict our attention on context-based offline meta-RL (COMRL), an understudied framework with a few existing algorithms (Li et al., 2019; Dorfman & Tamar, 2020; Mitchell et al., 2020; Li et al., 2021a), for a set of tasks that differ in reward or transition dynamics. One major challenge associated with this scenario is termed Markov Decision Process (MDP) ambiguity (Li et al., 2019; Dorfman & Tamar, 2020), namely the task-conditioned policies spuriously correlate task identity with state-action pairs due to biased distribution of the fixed datasets. This phenomenon can be interpreted as a special form of memorization problem in classical meta-learning (Yin et al., 2019), where the value and policy functions overfit the training distributions without capturing causality

---

[1]Source code is provided in the supplementary material.

from reward and transition functions, often leading to degenerate task representations (Li et al., 2021a) and poor generalization. To alleviate such over-fitting, Li et al. (2021a) proposes a framework named FOCAL which decouples the learning of task inference from control by using self-supervised distance metric learning. *However, they made a strong assumption on the existence of an injective map from each transistion tuple $\{s, a, s', r\}$ to its task identity.* Under extreme scenarios such as sparse reward, where a considerable portion of aggregated experience provides little information regarding task identity, efficient and robust learning of task representations is still challenging.

To address the aforementioned problem, in this paper we propose intra-task attention mechanism and inter-task contrastive learning objectives to achieve robust task inference. More specifically, for each task, we apply a batch-wise gated attention to recalibrate the weights of transition samples, and use sequence-wise self-attention (Vaswani et al., 2017b) to better capture the correlation within the transition (state, action, reward) dimensions. In addition, we implemented a matrix-form objective of the Momentum Contrast (MoCo) (He et al., 2020) for task-level representation learning, by replacing its dictionary queue with a meta-batch sampled on-the-fly. We provide theoretical analyses showing that our objective serves as a better surrogate than naive contrastive loss for task inference and the proposed attention mechanism on top can also reduce the variance of task representation. Moreover, empirical evaluations demonstrate that the proposed design choices of attention and contrastive learning mechanisms not only boost the performance of task inference, but also significantly improve its robustness against sparse reward and distribution shift. We name our new method FOCAL++.

## 2 RELATED WORK

**Attention in RL** Although attention mechanism has proven a powerful tool across of a broad spectrum of domains (Mnih et al., 2014; Vaswani et al., 2017a; Wang & Shen, 2017; Veličković et al., 2018; Devlin et al., 2018), to our best knowledge, its applications in RL remain relatively understudied. Most of previous works in RL (Mishra et al., 2018; Sukhbaatar et al., 2019; Kumar et al., 2020; Parisotto et al., 2020) focus on applying temporal attention in order to capture the time-dependent correlation in MDPs or POMDPs. Raileanu et al. (Raileanu et al., 2020) uses transformer as the default dynamics/policy encoder for meta-RL, similar to our proposed sequence-wise attention, without giving any intuition or comparative study on such design choice. Wang et al. (2021) recently implemented attention in meta-RL but didn't consider the offline setting.

The closest work we found by far (Barati & Chen, 2019; Li et al., 2021b) employ attention in multi-view/multi-agent RL, to learn different weights on various workers or agents, aggregated by a global network to form a centralized policy. Analogous to our proposal, such architecture has the advantage of adaptively accounting for inhomogeneous importance of each input in the decision making process, and makes the global agent robust to noise and partial observability.

**Contrastive Learning** Contrastive learning (Chopra et al., 2005; Hadsell et al., 2006) has emerged as a powerful framework for representation learning. In essence, it aims to capture data structures by learning to distinguish between semantically similar and dissimilar pairs. Recent progress in contrastive learning focuses mostly on learning visual representations as pretext tasks. MoCo (He et al., 2020) formulates contrastive learning as dictionary look-up, and builds a dynamic dictionary with a queue and a moving-averaged encoder. SimCLR (Chen et al., 2020) further pushes the SOTA benchmark with careful composition of data augmentations. However, all these algorithms concentrate primarily on generating pseudo-labels and contrastive pairs, whereas in COMRL scenario, the task labels and transition samples are naturally given.

There are a few recent works which apply contrastive learning in RL (Laskin et al., 2020) or meta-RL (Fu et al., 2020) settings. Fu et al. (2020) employs InfoNCE (Oord et al., 2018) loss to train a contrastive context encoder. They investigated the technique in the online setting, where the encoder requires an information-gain-based exploration strategy to be effective. In contrast, this paper focuses on how contrastive learning performs in the fully-offline setting.

**Context-Based Offline Meta-RL (COMRL)** Context-based offline meta-RL employs models with memory such as recurrent (Duan et al., 2016; Wang et al., 2016; Fakoor et al., 2020), recursive (Mishra et al., 2018) or probabilistic (Rakelly et al., 2019) structures to achieve fast adaptation by aggregating experience into a latent representation on which the policy is conditioned. To address the bootstrapping error problem (Kumar et al., 2019) for offline learning, framework like FOCAL

enforces behavior regularization (Wu et al., 2019), which constrains the distribution mismatch between the behavior and learning policies in actor-critic objectives. We follow the same paradigm.

## 3 METHOD

To tackle the COMRL problem, we follow the procedure described in FOCAL (Li et al., 2021a), by first learning an effective representation of tasks on latent space $\mathcal{Z}$, on which a single universal policy is conditioned and trained with behavior-regularized actor-critic method (Wu et al., 2019). As an improved version of FOCAL, our main contribution is twofold:

1. To our best knowledge, we are the first to apply attention mechanism in offline multi-task/meta-RL setting, for learning robust task representations. We combine batch-wise gated attention with sequence-wise transformer encoder, and demonstrate its lower variance as well as robustness against sparse reward and MDP ambiguity compared to prior COMRL methods.

2. On top of attention, we incorporate a matrix reformulation of Momentum Contrast (He et al., 2020) for task representation learning, with theoretical guarantees and provably better performance than ordinary contrastive objective.

### 3.1 PROBLEM SETUP

Consider a family of stationary MDPs defined by $\mathcal{M} = (\mathcal{S}, \mathcal{A}, \mathcal{P}, \mathcal{R}, \gamma)$ where $(\mathcal{S}, \mathcal{A}, \mathcal{P}, \mathcal{R}, \gamma)$ are the corresponding state space, action space, transition function, reward function and discount factor. A task $\mathcal{T}$ is defined as an instance of $\mathcal{M}$, which is associated with a pair of time-invariant transition and reward functions, $P(s'|s, a) \in \mathcal{P}$ and $R(s, a) \in \mathcal{R}$, respectively. In this work, we focus on tasks which share the same state and action space. Consequently, a task distribution can be modeled as a joint distribution of $\mathcal{P}$ and $\mathcal{R}$, which usually can be factorized:

$$p(\mathcal{T}) := p(\mathcal{P}, \mathcal{R}) = p(\mathcal{P})p(\mathcal{R}). \tag{1}$$

In the offline setting, each task $\mathcal{T}_i$ ($i$ being the task label) is associated with a static dataset of transition tuples $\mathcal{D}_i = \{c_i\} = \{(s_i, a_i, s'_i, R_i(s_i, a_i))\}$, for which $p(\mathcal{D}_i) = p(\mathcal{T}_i)$. Each tuple $c_i \sim \mathcal{D}_i$ is a sequence along the so-called *transition/sequence dimension*. A *meta-batch* $\mathcal{B}$ is a set of mini-batches $\mathcal{B}_i \sim \mathcal{D}_i$. Consider a meta-optimization objective in a multi-task form (Rakelly et al., 2019; Fakoor et al., 2020),

$$\mathcal{L}(\theta, \psi) = \mathbb{E}_{\mathcal{D}_i \sim p(\mathcal{D})}[\mathcal{L}_{\text{actor}}(\mathcal{D}_i; \theta) + \mathcal{L}_{\text{critic}}(\mathcal{D}_i; \psi)] \tag{2}$$

$$= \mathbb{E}_{\mathcal{D}_i \sim p(\mathcal{D})}[\mathcal{L}_{\mathcal{D}_i}(\theta, \psi)], \tag{3}$$

where $\mathcal{L}_{\mathcal{D}_i}(\theta, \psi)$ is the objective evaluated on transition samples drawn from $\mathcal{D}_i$, parameterized by $\theta$ and $\psi$. Assuming a common uniform distribution for a set of $n$ tasks, the meta-training procedure turns into minimizing the average losses across all training tasks

$$\hat{\theta}_{\text{meta}}, \hat{\psi}_{\text{meta}} = \arg \min_{\theta, \psi} \frac{1}{n} \sum_{k=1}^{n} \mathbb{E} \left[ \mathcal{L}_{\mathcal{D}_k}(\theta, \psi) \right]. \tag{4}$$

For COMRL problem, a task distribution corresponds to a family of MDPs on which a single universal policy is supposed to perform well. Since the MDP family is considered partially observed if no task identity information is given, a task inference module $E_\phi(z|c)$ is required to map context information $c \sim \mathcal{D}$ to a latent task representation $z \in \mathcal{Z}$ to form an augmented state, i.e.,

$$\mathcal{S}_{\text{aug}} \leftarrow \mathcal{S} \times \mathcal{Z}, \quad s_{\text{aug}} \leftarrow \text{concat}(s, z). \tag{5}$$

Such an MDP family is formalized as Task-Augmented MDP (TA-MDP) in FOCAL. Additionally, Li et al. (2021a) proves that a good task representation $z$ is crucial for optimization of the task-conditioned meta-objective in Eqn 4, which is the prime focus of this paper. We now show how to address the issue with the proposed attention architectures and contrastive learning framework.

### 3.2 ATTENTION ARCHITECTURES

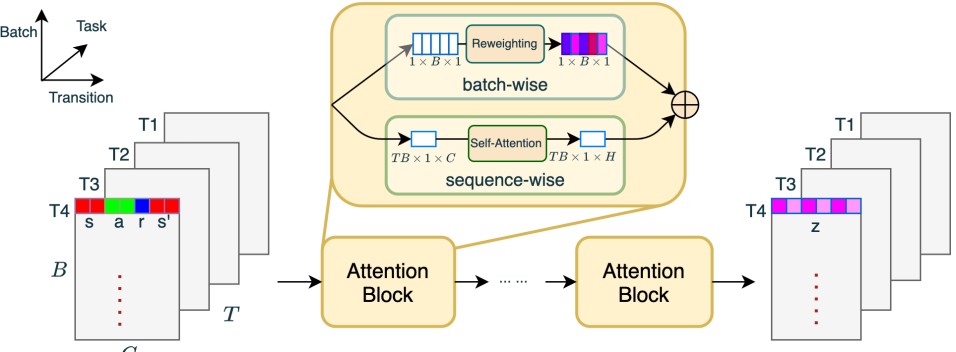

Figure 1: **Context encoder as a stack of attention blocks.**

We employ two forms of intra-task attention in the context encoder $E_\phi(\boldsymbol{z}|\boldsymbol{c})$: **batch-wise gated attention** and **sequence-wise self-attention**, for learning better task representations. The architectures are shown in Figure 2.

**Batch-Wise Gated Attention**   When performing task inference, transitions inside the same batch may contribute differently to the representation learning, especially in sparse reward situations. For tasks that differ in rewards, intuitively, transition samples with non-zero rewards contain more information regarding the task identity. Therefore, we utilize a gating mechanism similar to (Hu et al., 2018) along the batch dimension to adaptively recalibrates this batch-wise response by computing a scalar multiplier for every sample as in Figure 1.

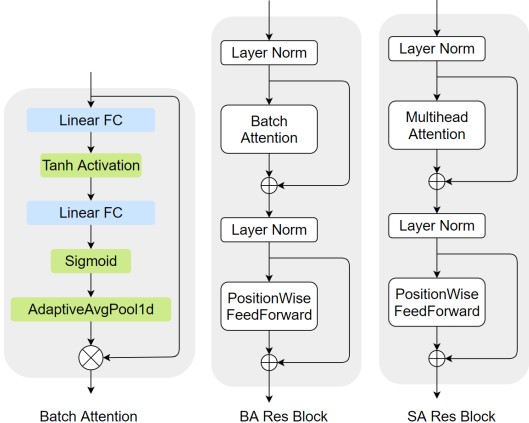

Figure 2: **Attention modules for task inference.** **BA**: batch-wise attention. **SA**: sequence-wise attention.

**Sequence-Wise Self-Attention**   A naive MLP encoder maps a concatenated 1-D sequence $(s, a, s', r)$ from context buffer to a 1-D embedding $\mathbf{z}$. This seq2seq model can be implemented with sequence-wise attention to apply self-attention along the sequence dimension. The intuition behind sequence-wise attention is that the attentive context encoder should in principle better capture the correlation in $(s, a, s', r)$ sequence related to task-specific reward function $R(s, a)$ and transition function $P(s'|s, a)$, compared to normal MLP layers employed by common context-based RL algorithms.

Illustrated in Figure 1, since two attention modules operate on separate dimensions, we connect them in parallel to generate task embedding $\mathbf{z}$ by addition.

Figure 3: **Inter-task matrix-form momentum contrast.** Given two meta-batches of transitions $\{\boldsymbol{c}^q\}$ and $\{\boldsymbol{c}^k\}$, a quickly progressing query encoder and a slowly progressing key encoder compute the corresponding batch-wise mean task representations in latent space $\mathcal{Z}$. A matrix multiplication is performed between the set of query and key vectors to produce the supervised contrastive loss in Eqn 8. $T, C, Z$ are the meta-batch, transition and latent space dimensions respectively.

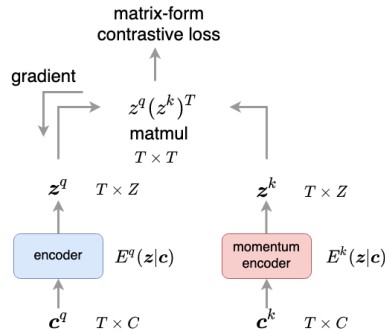

### 3.3 THE CONTRASTIVE LEARNING FRAMEWORK

Inspired by the successes of contrastive learning in computer vision (He et al., 2020), we process the raw transition data with momentum encoders to generate a latent query vector $\boldsymbol{z}^q$ as classifier and a set of $K$ latent key vectors $\{\boldsymbol{z}_0^k, \boldsymbol{z}_1^k, ..., \boldsymbol{z}_K^k\}$ as task representations. Suppose one of the keys $\boldsymbol{z}_+^k$ is the only match to $\boldsymbol{z}^q$, we employ the InfoNCE (Oord et al., 2018) objective as the building block:

$$\mathcal{L}_z = -\log \frac{\exp(\boldsymbol{z}^q \cdot \boldsymbol{z}_+^k / \tau)}{\sum_{i=0}^{K} \exp(\boldsymbol{z}^q \cdot \boldsymbol{z}_i^k / \tau)}, \tag{6}$$

where $\tau$ is a temperature hyper-parameter (Wu et al., 2018).

To ensure maximum sample efficiency, for each pair of meta-batches $\mathcal{B} = \{\mathcal{B}_i \sim \mathcal{D}_i | i = 1, ..., T\}$ where $T$ is the meta-batch size, one can construct $T$ InfoNCE objectives by taking the average latent vector of each task as the key, which is also crucial for our theoretical analysis (Theorem 3.1). Namely, given a meta-batch of encoded queries $\{\boldsymbol{z}_i^q \sim E_\phi^q(\boldsymbol{z}_i | \mathcal{B}_i) | i = 1, ..., T\}$ and keys $\{\boldsymbol{z}_i^k \sim E_\phi^k(\boldsymbol{z}_i | \mathcal{B}_i) | i = 1, ..., T\}$, our proposed contrastive loss is

$$\mathcal{L}_z = -\sum_{i=1}^{T} \log \frac{\exp(\boldsymbol{z}_i^q \cdot \boldsymbol{z}_i^k / \tau)}{\sum_{j=1}^{T} \exp(\boldsymbol{z}_i^q \cdot \boldsymbol{z}_j^k / \tau)}, \tag{7}$$

which can be written in a matrix-form

$$\mathcal{L}_z = -\operatorname{Tr}(M), \quad M_{ij} = \log \frac{\exp(\boldsymbol{z}_i^q \cdot \boldsymbol{z}_j^k / \tau)}{\sum_{j=1}^{T} \exp(\boldsymbol{z}_i^q \cdot \boldsymbol{z}_j^k / \tau)}. \tag{8}$$

The training scheme of our proposed inter-task momentum contrast is illustrated in Figure 3.

Now we provide a theoretical analysis of the objective in Eqn 8. Intuitively, it is the log loss of a $T$-way softmax-based classifier trying to classify each $\boldsymbol{z}_i^k$ as $\boldsymbol{z}_i^q$. With this interpretation, we compare it to a linear classifier with supervised loss and show that it can be recovered by the linear classifier if the weight matrix is a specific mean task classifier (Theorem 3.1). Furthermore, we prove that our proposed objective is a better surrogate than traditional contrastive loss for task inference.

**Definition 3.1 (Supervised Contrastive Loss)**

$$L_{sup}(\mathcal{T}, g) := \mathbb{E}_{\substack{\mathcal{T}_i, \mathcal{T}_{i'} \sim p(\mathcal{T}) \\ c_i \sim \mathcal{D}_i, c_{i'} \sim \mathcal{D}_{i'}}} [\ell(\{g(c_i) - g(c_{i'})\})]. \tag{9}$$

*where $\ell$ can be standard hinge or logistic losses as in (Saunshi et al., 2019).*

Consider a linear classifier $g(c) = \boldsymbol{W}E(c)$, where the encoded latent vector $E(c)$ is used as a deterministic representation (Li et al., 2021a) and $\boldsymbol{W} \in \mathbb{R}^{N \times Z}$ is a weight matrix trained to minimize $L_{sup}(\mathcal{T}, \boldsymbol{W}E)$, $Z$ is the dimension of the task latent space $\mathcal{Z}$. Such construction of contrastive objective enables self-supervised task representation learning for task inference, without requiring access to full labels of all possible tasks, which is flexible and has better potential for generalization. Hence the supervised loss of $E$ on $\mathcal{T}$ is defined as

$$L_{sup}(\mathcal{T}, E) = \inf_{\boldsymbol{W} \in \mathbb{R}^{N \times Z}} L_{sup}(\mathcal{T}, \boldsymbol{W}E). \tag{10}$$

Since the optimal $\boldsymbol{W}$ requires full knowledge and labels of the underlying task distribution, which is infeasible given only training tasks. As with Saunshi et al. (2019), we consider a particular choice of $\boldsymbol{W}^\mu$:

**Definition 3.2 (Mean Task Classifier)** *For an encoder function $E$ and a task set $\mathcal{T}$ of cardinality $N$, the **mean task classifier** $\boldsymbol{W}^\mu$ is an $N \times Z$ weight matrix whose $i^{th}$ row is the mean latent vector $\boldsymbol{\mu}_i$ of inputs with task label $i$. We use as a shorthand for its loss $L_{sup}^\mu(\mathcal{T}, E) := L_{sup}(\mathcal{T}, \boldsymbol{W}^\mu E)$.*

In pratice, we estimate the mean task representation of $\boldsymbol{z}^q$ and $\boldsymbol{z}^k$ using its batch-wise mean

$$\boldsymbol{\mu}_i^{q,k} := \mathbb{E}_{\substack{c_i \sim \mathcal{D}_i \\ \boldsymbol{z}_i^{q,k} \sim E_\phi^{q,k}(\boldsymbol{z}_i | c_i)}} [\boldsymbol{z}_i^{q,k}] \approx \mathbb{E}_{\substack{c_i \sim \mathcal{B}_i \\ \boldsymbol{z}_i^{q,k} \sim E_\phi^{q,k}(\boldsymbol{z}_i | c_i)}} [\boldsymbol{z}_i^{q,k}], \tag{11}$$

which induces the following definitions:

**Definition 3.3 (Averaged Supervised Contrastive Loss)** *Average supervised loss for an encoder function $E$ on $T$-way classification of task representation is defined as*

$$L_{sup}(E) := \mathop{\mathbb{E}}_{\{\mathcal{T}_i\}_{i=1}^T \sim p(\mathcal{T})} \left[ L_{sup}(\{\mathcal{T}_i\}_{i=1}^T, E) \right]. \tag{12}$$

*The average supervised loss of its mean classifier (Definition 3.2) is*

$$L_{sup}^{\mu}(E) := \mathop{\mathbb{E}}_{\{\mathcal{T}_i\}_{i=1}^T \sim p(\mathcal{T})} \left[ L_{sup}^{\mu}(\{\mathcal{T}_i\}_{i=1}^T, E) \right]. \tag{13}$$

When the loss function $\ell$ is the convex logistic loss, we prove in Appendix B that

**Theorem 3.1** *The matrix-form momentum contrast objective $\mathcal{L}_z$ (Eqn 8) is equivalent to the average supervised loss of its mean classifier $L_{sup}^{\mu}$ (Eqn 13) if $E = E^k(\boldsymbol{z}|\boldsymbol{c})$ is the key encoder and the mean task classifier $\boldsymbol{W}^{\mu}$ whose $i^{th}$ row is the mean of latent query vectors with task label $i$.*

If we compare our proposed loss function with the classical unsupervised contrastive loss

**Definition 3.4 (Unsupervised Contrastive Loss)**

$$L_{un}(E) := \mathbb{E} \left[ \ell(\{E(c)^T(E(c^+) - E(c^-))\}) \right]. \tag{14}$$

Given $T$ as the number of distinct tasks in meta-batches, $c, c^+$ are contexts from the same task, and $c^-$ is from the other $T - 1$ tasks. Such construction is employed by prior COMRL methods like FOCAL, which allows for task interpolation during meta-testing.

By **Lemma 4.3** in (Saunshi et al., 2019), using convexity of $\ell$ and Jensen's inequality, assuming no repeated task labels in each meta-batch, we have

**Theorem 3.2** *For all context encoder $E$*

$$L_{sup}(E) \leq L_{sup}^{\mu}(E) \leq L_{un}(E). \tag{15}$$

Combined with Theorem 3.1, it shows that our proposed contrastive objective in Eqn 8: $\mathcal{L}_z \equiv L_{sup}^{\mu}(E_\phi(\boldsymbol{z}|\boldsymbol{c}))$ serves as a better surrogate for $L_{sup}$ than the ordinary unsupervised contrastive losses employed by prior methods, to ensure similarity-preserving task representation for COMRL.

### 3.4 VARIANCE OF TASK EMBEDDINGS BY FOCAL++

In experiments, we found that our proposed algorithm, FOCAL++, which combines attention mechanism and matrix-form momentum contrast, exhibit significant smaller variance compared to the baselines on tasks with sparse reward (Table 2). We provide a proof of this observation for a simplified version of FOCAL++, by only considering the batch-wise attention along with contrastive learning objective defined in Eqn 8, *in presence of sparse reward. Assuming all tasks differ only in reward function*, we begin with the following definition:

**Definition 3.5 (Absolutely Sparse Transition)** *Given a set of tasks $\{T\}$ which only differ by reward function, a transition tuple (s,a,s',r) is **absolutely sparse** if $\forall \mathcal{T}_i \in \{\mathcal{T}\}, R_i(s, a) = constant$.*

According to policy invariance under reward transformations (Ng et al., 1999), without loss of generality, we assume the constant above to be zero for the rest of the paper.

**Definition 3.6 (Task with Sparse Reward)** *For a dataset $\mathcal{D}_i = \{(s_i, a_i, s_i', R_i(s_i, a_i))\}$ sampled from any task $\mathcal{T}_i$ with sparse reward, it can be decomposed as a disjoint union of two sets of transitions:*

$$\mathcal{D}_i = \{(s_i, a_i, s_i', R_i(s_i, a_i))\} \cup \{(s_i, a_i, s_i', 0)\} \tag{16}$$
$$= \{c_n\} \cup \{c_s\}, \tag{17}$$

where $\{c_s\}$ is the set of absolutely sparse transitions (Definition 3.5), which by definition are shared across all tasks. $\{c_n\}$ consists of the rest of the transitions, and is unique to task $\mathcal{T}_i$.

**Definition 3.7 (Batch-Wise Gated Attention)** *The batch-wise gated attention assigns inhomogeneous weights $\boldsymbol{W}$ for batch-wise estimation of the mean task representation of $\boldsymbol{\mu}^{q,k}$ in Eqn 11:*

$$\boldsymbol{\mu}_i^{q,k}(\boldsymbol{W}) := \mathbb{E}_{c\sim\mathcal{D}_i}[\boldsymbol{W}(c)E^{q,k}(c)] \tag{18}$$

$$= p_n\mathbb{E}[\boldsymbol{W}(c_n)E^{q,k}(c_n)] + p_s\mathbb{E}[\boldsymbol{W}(c_s)E^{q,k}(c_s)], \tag{19}$$

where $p_n, p_s$ are the measures of $\{c_n\}, \{c_s\}$ respectively and $\boldsymbol{W}$ is normalized such that $\mathbb{E}_{c\sim\mathcal{D}_i}[\boldsymbol{W}(c)] = 1$. $p_n + p_s = 1$ by Definition 3.6.

**Theorem 3.3** *Given a learned batch-wise gated attention weight $\boldsymbol{W}$ and context encoder $E$ that minimize the contrastive learning objective $L_{sup}^{\mu}(\boldsymbol{W}, E)$, we have*

$$Var(\mu_i^{q,k}(\boldsymbol{W})) \leq Var(\boldsymbol{\mu}_i^{q,k}), \tag{20}$$

*when the sparsity ratio exceeds a threshold.*

i.e., the variance of learned task embeddings with batch attention is upper-bounded by its counterpart without attention given the dataset is sparse enough. We prove Theorem 3.3 in Appendix B.

## 4 EXPERIMENTS

In the following experiments, we show FOCAL++ outperforms the existing COMRL algorithms by a clear margin in three key aspects: a) asymptotic performance of learned policy; b) task representations with lower variance; and c) robustness to sparse reward and MDP ambiguity.

All trials are averaged over 3 random seeds. The offline training data are generated in accordance with the protocol of FOCAL by training stochastic SAC (Haarnoja et al., 2018) models for every distinct task and roll out policies saved at each checkpoint to collect trajectories. The offline training datasets can be collected as a selection of the saved trajectories, which facilitates tuning of the performance level and state-action distributions (Table 3). Both training and testing sets are pre-collected, making our method fully-offline. *Rewards are sparsified by constructing a neighborhood of goal in state or velocity space, where transition samples which lie outside the area are assigned zero reward. Since the focus of this paper is robust task representation learning which can be decoupled from control according to FOCAL, we use sparse-reward data only when training the context encoders. Learning of meta-policy in presence of sparse reward is another active but orthogonal area of research where quite a few successful solutions have been found (Andrychowicz et al., 2017; Eysenbach et al., 2020).* A concrete description of the hyper-parameters and experimental settings is covered in Appendix D.

Table 1: Average testing return (standard deviation in parenthesis) of FOCAL and variants of FOCAL++.

| Algorithm | Sparse-Point-Robot | Point-Robot-Wind | Sparse-Cheetah-Dir | Sparse-Ant-Dir | Sparse-Cheetah-Vel | Walker-2D-Params |
|---|---|---|---|---|---|---|
| FOCAL | $11.84_{(1.05)}$ | $-5.61_{(0.59)}$ | $1351.40_{(90.46)}$ | $429.92_{(41.52)}$ | $-183.32_{(40.16)}$ | $302.70_{(12.94)}$ |
| FOCAL++ (contrastive) | $12.53_{(0.31)}$ | $-5.78_{(0.44)}$ | $1309.76_{(115.33)}$ | $504.00_{(145.80)}$ | $-158.95_{(21.36)}$ | $366.35_{(55.08)}$ |
| FOCAL++ (batch-wise) | $12.54_{(0.23)}$ | $-5.57_{(0.34)}$ | $1330.56_{(162.03)}$ | $687.37_{(85.95)}$ | $-150.58_{(11.75)}$ | $376.52_{(36.59)}$ |
| FOCAL++ (seq-wise) | $12.64_{(0.14)}$ | $\mathbf{-5.09}_{(\mathbf{0.01})}$ | $1293.40_{(129.99)}$ | $573.26_{(186.22)}$ | $-140.63_{(11.52)}$ | $375.67_{(45.72)}$ |
| FOCAL++ | $\mathbf{12.96}_{(\mathbf{0.09})}$ | $-5.39_{(0.57)}$ | $\mathbf{1470.52}_{(\mathbf{68.29})}$ | $\mathbf{719.77}_{(\mathbf{57.58})}$ | $\mathbf{-137.31}_{(\mathbf{7.06})}$ | $\mathbf{391.02}_{(\mathbf{42.44})}$ |

Table 2: Variance of context embeddings averaged over all training tasks and latent dimensions.

| Algorithm | Sparse-Point-Robot | Point-Robot-Wind | Sparse-Cheetah-Dir | Sparse-Ant-Dir | Sparse-Cheetah-Vel | Walker-2D-Params |
|---|---|---|---|---|---|---|
| FOCAL | 8.54E-5 | 3.05E-3 | 4.31E-3 | 2.24E-3 | 2.57E-3 | 1.06E-2 |
| FOCAL++ (contrastive) | 7.83E-5 | 1.68E-3 | 6.86E-4 | 1.77E-3 | 1.73E-3 | 5.79E-3 |
| FOCAL++ (batch-wise) | **7.73E-5** | 1.70E-3 | **4.66E-4** | **7.51E-4** | 1.04E-3 | 5.85E-3 |
| FOCAL++ (seq-wise) | 7.94E-5 | 1.84E-3 | 9.43E-4 | 8.00E-4 | **9.76E-4** | 5.46E-3 |
| FOCAL++ | 8.27E-5 | **1.68E-3** | 7.82E-4 | 1.35E-3 | 1.06E-3 | **5.23E-3** |

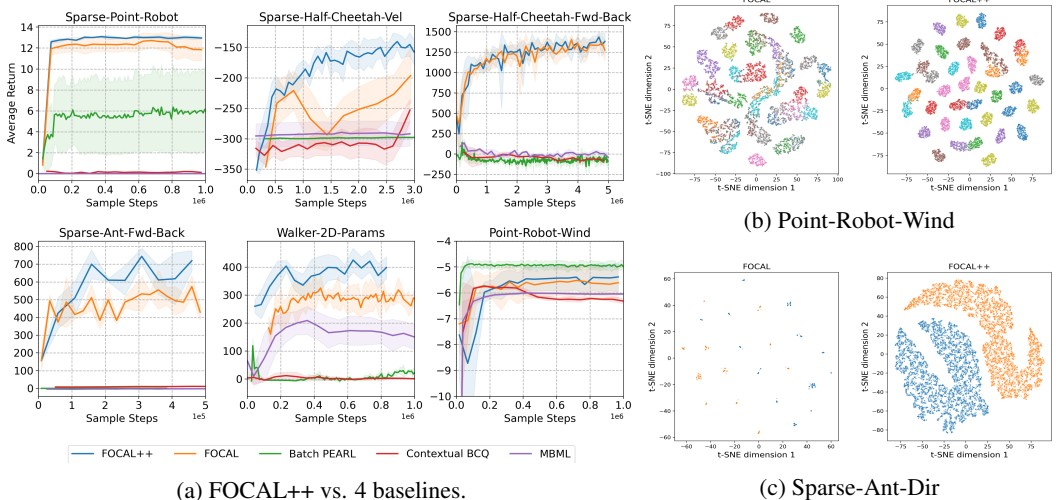

Figure 4: **Left:** Test-task performance vs. transition steps sampled for meta-training. **Right:** t-SNE visualization of the learned task embeddings $z^q$ on Point-Robot-Wind and Sparse-Ant-Dir. Each point represents a query vector which is color-coded according to its task label.

## 4.1 ASYMPTOTIC PERFORMANCE

We evaluate FOCAL++ on 6 continuous control meta-environments of robotic locomotion (Todorov et al., 2012) adopted from FOCAL. 4 (Sparse-Point-Robot, Sparse-Cheetah-Vel, Sparse-Cheetah-Fwd-Back, Sparse-Ant-Fwd-Back) and 2 (Point-Robot-Wind, Walker-2D-Params) environments require adaptation by reward and transition functions respectively. For inference, FOCAL++ aggregates context from a fixed test set to infer task embedding, and is subsequently evaluated online. Besides FOCAL, three other baselines are compared: an offline variant of the PEARL algorithm (Rakelly et al., 2019) (Batch PEARL), a context-based offline BCQ algorithm (Fujimoto et al., 2019) (Contextual BCQ) and a two-stage COMRL algorithm with reward/dynamics relabelling (Li et al., 2019) (MBML).

Shown in Figure 4a, FOCAL outperforms other methods across almost all domains with context embeddings of higher quality in Figure 4b,4c. In Table 1, our ablation studies also show that each design choice of FOCAL++ alone can improve the performance of the learned policy, and combining the orthogonal intra-task attention mechanism with inter-task contrastive learning yields the best outcome.

## 4.2 ROBUSTNESS TO MDP AMBIGUITY AND SPARSE REWARD

In our experimental setup, an ideal context encoder should capture the generalizable information for task inference, namely the difference between reward/dynamics functions across a distribution of tasks. However, as discussed in Section 1, there are two major challenges that impede conventional COMRL algorithms from learning robust representations:

Table 3: Average testing return of FOCAL and FOCAL++ on Sparse-Point-Robot with different distributions of training/testing sets. The numbers in parenthesis represent performance drop due to distribution shift. Additional experiments are presented in Apppendix C.

| Environment | Training | Testing | FOCAL | FOCAL++ |
|---|---|---|---|---|
| Sparse-Point-Robot | expert | expert | 8.16 | 12.60 |
| | | medium | $7.12_{(1.04)}$ | $12.47_{(0.13)}$ |
| | | random | $4.43_{(3.73)}$ | $10.17_{(2.43)}$ |
| | medium | medium | 8.44 | 12.54 |
| | | expert | $8.25_{(0.19)}$ | $12.44_{(0.10)}$ |
| | | random | $6.76_{(1.68)}$ | $10.49_{(2.05)}$ |
| Walker-2D-Params | mixed | mixed | 302.70 | 391.02 |
| | | expert | $271.69_{(31.01)}$ | $377.46_{(13.56)}$ |

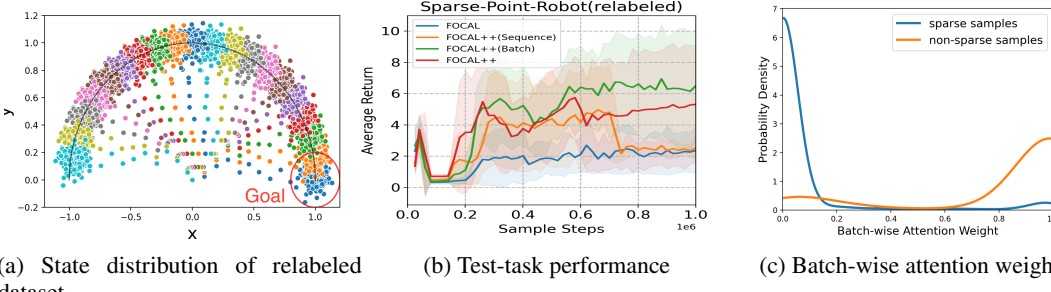

(a) State distribution of relabeled dataset

(b) Test-task performance

(c) Batch-wise attention weight

Figure 5: **Result on the relabeled Sparse-Point-Robot dataset**. (a) State distributions of the expert datasets for 20 distinct tasks, with goals uniformly distributed on a semicircle. (b) On mixed dataset, FOCAL completely fails in this scenario whereas FOCAL++ variants with batch-wise attention are able to learn. (c) Probability distribution of the batch-wise attention weight of samples with absolutely zero and non-zero reward. Binary classification AUC = 0.969.

**MDP ambiguity** arises due to COMRL algorithms' sensitivity to fixed dataset distributions (Li et al., 2019; Dorfman & Tamar, 2020). Take Sparse-Point-Robot for example, as in Figure 5a, for tasks with a goal on the semicircle, the state-action distribution exhibits specific pattern which may reflect task identity. Given $\mathcal{D} = \{(s, a, s', r)\}$ as input, the context encoder may learn a spurious correlation between state-action distributions and task identity, which causes performance degradation under distribution shifts (Table 3).

**Sparse reward** in meta-environments could exacerbate MDP ambiguity by making a considerable portion of transitions uninformative for task inference, such as the samples outside any goals in Figure 5a. Attention mechanism, especially the batch-wise channel attention, helps the context encoder attend to the informative portion of the input transitions, and therefore significantly improve the robustness of the learned policies.

To demonstrate the robustness of FOCAL++ in presence of the two challenges above, we tested it against distribution shift by using datasets of various qualities: expert, medium, random and mixed which combines all three. Shown in Table 3, we observe that overall the performance drop due to distribution shift is significantly lower when attention and contrastive learning are applied.

Moreover, we are aware that even mixing of datasets generated by different behavior policies cannot fully eliminate the risk of MDP ambiguity since the state-action distributions for each task still do not completely overlap. To show that the attention modules introduced by FOCAL++ indeed works as intended by capturing the reward-task dependency, we create a new dataset on Sparse-Point-Robot by merging the state-action support across all tasks and relabelling the sparse reward according to the task-specific reward functions. In principle, this fully prevents information leakage from the state-action distributions, forcing the context encoder to learn to distinguish the reward functions between tasks while minimizing the contrastive loss. Shown in Figure 5b, we experimented with 3 attention variants of FOCAL++ on the relabeled dataset, and found that batch-wise attention significantly improves the performance as intended. Additionally, we visualize the density distribution of batch-wise attention weights assigned to samples in Figure 5c. We see a clear tendency for the batch-attention module to assign zero weight to samples with zero rewards (the absolutely sparse data points which lie outside all goal circles in Figure 5a) and maximum weights to the non-zero-reward transitions, with binary classification AUC = 0.969, which is clear evidence of FOCAL++ learning the correct correlation for task inference by attending to the informative context.

## 5 CONCLUSION

In this work, we address the understudied COMRL problem and provably improve upon the existing SOTA baselines such as FOCAL, by focusing on more effective and robust learning of task representations. Key to our framework is the combination of intra-task attention mechanism and inter-task contrastive learning, for which we provide theoretical grounding and experimental evidence on the superiority of our design.

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

## APPENDIX A   PSEUDO-CODE[2]

---

**Algorithm 1:** FOCAL++ Meta-training

- Pre-collected batch $\mathcal{D}_i = \{(s_j, a_j, s'_j, r_j)\}_{j:1,\ldots,N}$ from a set of training tasks $\{\mathcal{T}_i\}_{i=1,\ldots,n}$ drawn from $p(\mathcal{T})$
- Learning rates $\alpha_1, \alpha_2, \alpha_3$, temperature $\tau$, momentum $m$

1  Initialize context replay buffer $\mathcal{C}_i$ for each task $\mathcal{T}_i$
2  Initialize context encoder network $E_\phi^{q,k}(z|c)$, learning policy $\pi_\theta(a|s,z)$ and Q-network $Q_\psi(s,z,a)$ with parameters $\phi_q$, $\phi_k$, $\theta$ and $\psi$
3  **while** not done **do**
4      **for** each $\mathcal{T}_i$ **do**
5          **for** t = 0, $T-1$ **do**
6              Sample mini-batches of B transitions $\{(s_{i,t}, a_{i,t}, s'_{i,t}, r_{i,t})\}_{t:1,\ldots,B} \sim \mathcal{D}_i$ and update $\mathcal{C}_i$
7          **end**
8      **end**
9      Sample a pair of query-key meta-batches of $T$ tasks $\sim p(\mathcal{T})$
10     **for** step in training steps **do**
11         **for** each $\mathcal{T}_i$ **do**
12             Sample mini-batches $c_i$ and $b_i \sim \mathcal{C}_i$ for context encoder and policy training ($b_i$, $c_i$ are identical by default, the rewards in $b_i$ are always **non-sparse**)
13             Compute $\boldsymbol{z}_i^q = E_\phi^q(c_i)$
14             **for** each $\mathcal{T}_j$ **do**
15                 Sample mini-batches $c_j$ from $\mathcal{C}_j$ and compute $\boldsymbol{z}_j^k = E_\phi^k(c_j)$
16                 $\mathcal{M}_{ij} = \mathcal{M}_z(\boldsymbol{z}_i^q, \boldsymbol{z}_j^k)$         ▷ **matrix-form momentum contrast**
17             **end**
18             $\mathcal{L}_{actor}^i = \mathcal{L}_{actor}(b_i, E_\phi^q(c_i))$
19             $\mathcal{L}_{critic}^i = \mathcal{L}_{critic}(b_i, E_\phi^q(c_i))$
20         **end**
21         $\mathcal{L}_z = \text{Tr}(M)$
22         $\phi_q \leftarrow \phi_q - \alpha_1 \nabla_{\phi_q} \mathcal{L}_z$
23         $\phi_k \leftarrow m\phi_k + (1-m)\phi_q$         ▷ **momentum update**
24         $\theta \leftarrow \theta - \alpha_2 \nabla_\theta \sum_i \mathcal{L}_{actor}^i$
25         $\psi \leftarrow \psi - \alpha_3 \nabla_\psi \sum_i \mathcal{L}_{critic}^i$
26     **end**
27 **end**

---

**Algorithm 2:** FOCAL++ Meta-testing

- Pre-collected batch $\mathcal{D}_{i'} = \{(s_{j'}, a_{j'}, s'_{j'}, r_{j'})\}_{j':1,\ldots,M}$ from a set of testing tasks $\{\mathcal{T}_{i'}\}_{i'=1\ldots m}$ drawn from $p(\mathcal{T})$

1  Initialize context replay buffer $\mathcal{C}_{i'}$ for each task $\mathcal{T}_i$
2  **for** each $\mathcal{T}_{i'}$ **do**
3      **for** t = 0, $T-1$ **do**
4          Sample mini-batches of B transitions $c_{i'} = \{(s_{i',t}, a_{i',t}, s'_{i',t}, r_{i',t})\}_{t:1,\ldots,B} \sim \mathcal{D}_{i'}$ and update $\mathcal{C}_{i'}$
5          Compute $z_{i'}^q = E_\phi^q(c_{i'})$
6          Roll out policy $\pi_\theta(a|s, z_{i'}^q)$ for evaluation
7      **end**
8  **end**

---

[2]To prevent conflict or misunderstanding, all non-hyperlink equation/theorem numbers in the appendix refer to those in the main text.

## APPENDIX B    DEFINITIONS AND PROOFS

### B.1    PROOF OF THEOREM 3.1

Consider a task set $\{\mathcal{T}\} = \{\mathcal{T}_1, ..., \mathcal{T}_T\}$ drawn uniformly from $p(\mathcal{T})$. In Definition 3.1, the loss incurred by $g$ on point $(c, \mathcal{T}_i) \in \mathcal{C} \times \{\mathcal{T}\}^3$ is defined as $\ell(\{g(c)_i - g(c)_{i'}\}_{i' \neq i})$, which is a function of a $T$-dimensional vector of differences in the coordinates. Given the definition of the mean task classifier $\boldsymbol{W}^\mu$ that $g(c) = \boldsymbol{W}E(c)$ and $\ell(\boldsymbol{v}) = \log(1 + \sum_i \exp(-v_i))$ the standard logistic loss as in (Saunshi et al., 2019), the supervised contrastive loss defined in Eqn 10 can be rewritten as

$$L_{sup}(\mathcal{T}, g) := \mathbb{E}_{\substack{\mathcal{T}_i \sim p(\mathcal{T}) \\ c_i \sim \mathcal{D}}} \left[ \log \left( 1 + \sum_{i' \neq i} \exp \left( \sum_j (\boldsymbol{W}_{i'j}E(c_i)_j - \boldsymbol{W}_{ij}E(c_i)_j) \right) \right) \right]. \quad (21)$$

Since the $i^{th}$ row of $\boldsymbol{W}$ is the mean of latent key vectors with task label $i$, and $E = E^k(\boldsymbol{z}|\boldsymbol{c})$ is the key encoder, Eqn 21 turns into

$$L_{sup}(\mathcal{T}, g) := \mathbb{E}_{\mathcal{T}_i \sim p(\mathcal{T})} \left[ \log \left( 1 + \sum_{i' \neq i} \exp \left( \boldsymbol{z}_{i'}^k \cdot \boldsymbol{z}_i^q - \boldsymbol{z}_i^k \cdot \boldsymbol{z}_i^q \right) \right) \right]. \quad (22)$$

In practice, we estimate the latent vectors $\boldsymbol{z}_i^{q,k}$ using batch-wise mean to approximate the the mean task representation $\mu_i^{q,k}$. Therefore $L_{sup}$ in 22 is equivalent to the mean task classifier $L_{sup}^\mu$ defined in Definition 3.2. One step futher, assuming uniform distribution of the task set $\{\mathcal{T}\}^4$, the averaged supervised contrastive loss by Definition 3.3 is

$$\begin{aligned} L_{sup}^\mu(E) &:= \mathbb{E}_{\{\mathcal{T}_i\}_{i=1}^T \sim p(\mathcal{T})} \left[ L_{sup}^\mu(\{\mathcal{T}_i\}_{i=1}^T, E) \right] \\ &= \frac{1}{T} \sum_{i=1}^T \left[ \log \left( 1 + \sum_{i' \neq i} \exp \left( \boldsymbol{z}_{i'}^k \cdot \boldsymbol{z}_i^q - \boldsymbol{z}_i^k \cdot \boldsymbol{z}_i^q \right) \right) \right] \quad (23) \\ &= -\frac{1}{T} \sum_{i=1}^T \log \frac{\exp \left( \boldsymbol{z}_i^q \cdot \boldsymbol{z}_i^k \right)}{\sum_{j=1}^T \exp \left( \boldsymbol{z}_i^q \cdot \boldsymbol{z}_j^k \right)}, \quad (24) \end{aligned}$$

which is precisely the matrix-form momentum contrast objective (Eqn 8,9) if one rescales $\boldsymbol{W}$ by a factor of $\tau$.

### B.2    PROOF OF THEOREM 3.3

With Definition 3.5, 3.6 and 3.7, we hereby provide a simplified proof by assuming a constant weight $\boldsymbol{W}(c)$ on the non-sparse set $\{c_n\}$ and the absolutely sparse set $\{c_s\}$ (Definition 3.5) respectively, then we have

$$\mu_i^{q,k}(\boldsymbol{W}) = p_n \boldsymbol{W}(c_n) \mathbb{E}_{c_n \sim \{c_n\}}[E^{q,k}(c_n)] + p_s \boldsymbol{W}(c_s) \mathbb{E}_{c_s \sim \{c_s\}}[E^{q,k}(c_s)], \quad (25)$$

where the normalization condition $\mathbb{E}_{c \sim \mathcal{D}_i}[\boldsymbol{W}(c)] = 1$ implies $p_n \boldsymbol{W}(c_n) + p_s \boldsymbol{W}(c_s) = 1$. Therefore, adding the batch-wise attention is effectively modulating $p_n$ and $p_s$. Since $p_n + p_s = 1$, without loss of generality, we apply the following notations:

---

[3] $\mathcal{C} = \{(s_i, a_i, s_i', R_i(s_i, a_i))\}$ is the context space

[4] **Note that the task set $\{\mathcal{T}\}$ discussed here is a *subset* of the whole task set and does not necessarily cover the whole support of $p(\mathcal{T})$. It is sampled for the sole purpose of computing the contrastive loss.**

$$p_n = p, \quad p_n \boldsymbol{W}(c_n) = p' \tag{26}$$

$$\mathbb{E}_{c_n \sim \{c_n\}}[E^{q,k}(c_n)] = \boldsymbol{x}_n^{q,k}, \quad \mathbb{E}_{c_s \sim \{c_s\}}[E^{q,k}(c_s)] = \boldsymbol{x}_s^{q,k}. \tag{27}$$

Assuming i.i.d $\boldsymbol{x}_n$ and $\boldsymbol{x}_s$, which gives

$$\text{Var}(\mu_i^{q,k}(\boldsymbol{W})) = \text{Var}(p'\boldsymbol{x}_n^{q,k} + (1-p')\boldsymbol{x}_s^{q,k}) = (p')^2\text{Var}(\boldsymbol{x}_n^{q,k}) + (1-p')^2\text{Var}(\boldsymbol{x}_s^{q,k}) \tag{28}$$

$$\text{Var}(\boldsymbol{\mu}_i^{q,k}) = \text{Var}(p\boldsymbol{x}_n^{q,k} + (1-p)\boldsymbol{x}_s^{q,k}) = p^2\text{Var}(\boldsymbol{x}_n^{q,k}) + (1-p)^2\text{Var}(\boldsymbol{x}_s^{q,k}). \tag{29}$$

By B.1, the averaged supervised loss $L_{sup}^\mu(\boldsymbol{W}, E)$ is equivalent to the matrix-form contrastive objective, which can be written as

$$L_{sup}^\mu(\boldsymbol{W}, E) = \frac{1}{T}\sum_{i=1}^T \left[\log\left(1 + \sum_{i' \neq i} \exp\left((\boldsymbol{\mu}_{i'}^k - \boldsymbol{\mu}_i^k) \cdot \boldsymbol{\mu}_i^q\right)\right)\right]$$

$$= \frac{1}{T}\sum_{i=1}^T \left[\log\left(1 + \sum_{i' \neq i} \exp\left(p'(\boldsymbol{x}_{i'}^k - \boldsymbol{x}_i^k) \cdot \boldsymbol{\mu}_i^q\right)\right)\right], \tag{30}$$

where we use the definition of $\boldsymbol{\mu}$ in Eqn 25 and the fact that $x_s^{q,k}$ is the same across all tasks. Since the learned $\widehat{\boldsymbol{W}}, \widehat{E} \in \arg\min_{\boldsymbol{W} \in \mathcal{A}, E \in \mathcal{E}} L_{sup}^\mu(\boldsymbol{W}, E)$, and $p' \approx p$ by the identity map initialization of the *residual* attention module, we have, for learned $\widehat{p'}, \widehat{\boldsymbol{x}}$ and $\widehat{\boldsymbol{\mu}}$,

$$\widehat{p'} \geq p, \quad (\widehat{\boldsymbol{x}}_{i'}^k - \widehat{\boldsymbol{x}}_i^k) \cdot \widehat{\boldsymbol{\mu}}_i^q < 0. \tag{31}$$

Now subtract Eqn 28 by 29, we have

$$\text{Var}(\mu_i^{q,k}(\widehat{\boldsymbol{W}})) - \text{Var}(\boldsymbol{\mu}_i^{q,k}) = [(\widehat{p'})^2 - p^2]\text{Var}(\boldsymbol{x}_n^{q,k}) + [(1-\widehat{p'})^2 - (1-p)^2]\text{Var}(\boldsymbol{x}_s^{q,k})$$

$$= (\widehat{p'} - p)\left[(\widehat{p'} + p)\text{Var}(\boldsymbol{x}_n^{q,k}) - (2 - p - \widehat{p'})\text{Var}(\boldsymbol{x}_s^{q,k})\right]$$

$$\leq 0, \quad if \quad p \leq \widehat{p'} \leq \frac{(2-p)\text{Var}(\boldsymbol{x}_s^{q,k}) - p\text{Var}(\boldsymbol{x}_n^{q,k})}{\text{Var}(\boldsymbol{x}_n^{q,k}) + \text{Var}(\boldsymbol{x}_s^{q,k})}. \tag{32}$$

The left inequality automatically holds by Eqn 31, the RHS is satisfied when

$$p \leq \frac{\text{Var}(\boldsymbol{x}_s^{q,k})}{\text{Var}(\boldsymbol{x}_n^{q,k}) + \text{Var}(\boldsymbol{x}_s^{q,k})}, \tag{33}$$

or equivalently,

$$p_s = (1-p) \geq \frac{\text{Var}(\boldsymbol{x}_n^{q,k})}{\text{Var}(\boldsymbol{x}_n^{q,k}) + \text{Var}(\boldsymbol{x}_s^{q,k})}, \tag{34}$$

which means when the sparsity of reward exceeds the threshold, a learned batch attention module can reduce the variance of the mean task representation $\boldsymbol{\mu}_i^{q,k}$. Eqn 34 is corroborated by our experiments on the relabeled Sparse-Point-Robot dataset (Figure 6).

## APPENDIX C    ADDITIONAL EXPERIMENTS

In Table 4, we present more experimental evidence that FOCAL++ is more robust against distribution shift compared to FOCAL on Walker-2D-Params, which is consistent with Table 3 in the main text.

Table 4: Extension of Table 3 in the main text. Average testing return of FOCAL and FOCAL++ for more settings of distribution shift on Walker-2D-Params.

| Environment | Training | Testing | FOCAL | FOCAL++ |
|---|---|---|---|---|
| Walker-2D-Params | expert | expert | 373.92 | 364.75 |
| | | mixed | $322.24_{(51.68)}$ | $340.60_{(24.15)}$ |
| | | random | $284.94_{(88.98)}$ | $297.43_{(67.32)}$ |
| | mixed | mixed | 302.70 | 391.02 |
| | | expert | $271.69_{(31.01)}$ | $377.46_{(13.56)}$ |
| | | random | $260.02_{(42.68)}$ | $346.95_{(44.07)}$ |

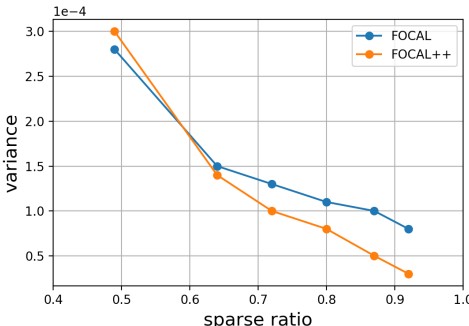

Figure 6: The variance-sparsity relation for FOCAL++/FOCAL on the relabeled Sparse-Point-Robot dataset. The y-axis measures the variance of the bounded task embeddings $z \in (-1, 1)^l$ averaged over all $l$ latent dimensions. See more details in D.2.

Moreover, to testify our conclusion in B.2, we present the variance of task embedding vectors of FOCAL++ and FOCAL under various sparsity levels. Shown in Figure 6, the variance of the weighted embeddings $\boldsymbol{\mu}_i^{q,k}(\hat{\boldsymbol{W}})$ becomes lower than its unweighted counterpart $\boldsymbol{\mu}_i^{q,k}$ when sparse ratio exceeds a threshold about $0.6$. The observation matches well with Eqn 34 we derived in B.2.

## APPENDIX D    EXPERIMENTAL DETAILS AND HYPERPARAMETER

### D.1    OVERVIEW OF THE META ENVIRONMENTS

The meta-environments could be divided into two categories: meta-environments that *only differ in reward function* and that *only differ in transition function*. For the meta-environments that *only differ in reward functions*, we additionally introduce sparsity to the reward function.

- **Sparse-Point-Robot** is a 2D-navigation task with sparse reward, introduced in Rakelly et al. (2019). Each task is associated with a goal sampled uniformly on a unit semicircle. The agent is trained to navigate to set of goals, then tested on a distinct set of unseen test goals. Tasks differ in *reward* function only.

- **Point-Robot-Wind** is another variant of Sparse-Point-Robot. Each task is associated with the same reward but a distinct "wind" sampled uniformly from $[-l, l]^2$. Every time the agent takes a step, it drifts by the wind vector. We set $l = 0.05$ in this paper. Tasks differ in *transition* function only.

- **Sparse-Cheetah-Vel, Sparse-Ant-Fwd-Back, Sparse-Cheetah-Fwd-Back** are sparse-reward variants of the popular meta-RL benchmarks Half-Cheetah-Vel, Sparse-Ant-Dir and Sparse-Cheetah-Fwd-Back based on MuJoCo environments, introduced by Finn et al. (2017) and Rothfuss et al. (2018). Tasks differ in *reward* function only.

- **Walker-2D-Params** is a unique environment compared to other MuJoCo environments. Agent is initialized with some system dynamics parameters randomized and must move forward. Transitions function is dependent on randomized task-specific parameters such as mass, inertia and friction coefficients. Tasks differ in *transition* function only.

The way we sparsify the reward functions is as follows.

$$\text{sparsified reward} = \begin{cases} \frac{\text{reward}-\text{goal radius}}{|\text{ goal radius }|}, & \text{if reward} > \text{goal radius} \\ 0, & \text{otherwise} . \end{cases} \tag{35}$$

Intuitively, we set rewards of states that lie outside a neighborhood of the goal to 0, and re-scaled the rewards otherwise so that the sparse reward function is continuous. For each of the sparsified environments other than the relabeled Sparse-Point-Robot, we set its goal radius to achieve a non-sparse rate of about 50%. *Note that only the transitions used for training the context-encoder are sparsified,* since the focus of this paper is learning effective and robust task representations.

### D.2    RELABELED DATASET

As discussed in Section 4.3, to prevent information leakage of task identity from state-action distribution, we construct the **relabeled Sparse-Point-Robot dataset** from a pre-collected dataset of the Sparse-Point-Robot environment.

Figure 7 illustrates the generating process for task 2 of the original dataset. The original state distribution of five example tasks on Sparse-Point-Robot is shown in the upper-left. After merging the transition state-action support across all tasks, the (state, action, next state) distribution are identical for every specific task. Then we recompute the reward for each transition according to the task-specific reward functions and sparsify the result. We perform the merge-relabel-sparsify process for all tasks on Sparse-Point-Robot to enhance the importance of the non-sparse samples for task inference. *The sparse samples in Figure 7 of the main text are those that lie outside of all goals, i.e. transitions with zero reward across all tasks.*

The dataset can be accessed and downloaded from relabeled_dataset.

### D.3    HYPERPARAMETERS

Tables 5 and 6 describe the hyperparameters used in our empirical evaluations.

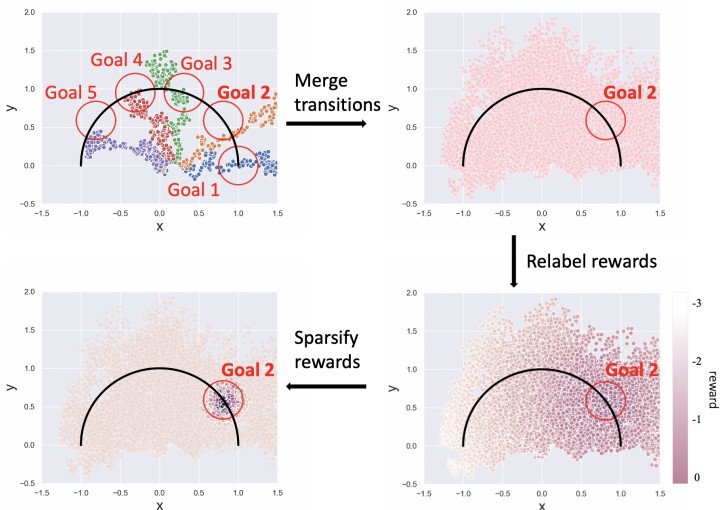

Figure 7: Generating process of the relabeled Sparse-Point-Robot dataset.

Table 5: Specifications of the environments experimented in our paper.

| Training Set | Training Tasks | Testing Tasks | Goal Radius |
|---|---|---|---|
| Sparse-Point-Robot | 80 | 20 | -0.2 |
| Sparse-Point-Robot (relabeled) | 80 | 20 | -0.5 |
| Point-Robot-Wind | 40 | 10 | N/A |
| Sparse-Cheetah-Vel | 80 | 20 | -0.1 |
| Sparse-Ant-Fwd-Back | 2 | 2 | 3 |
| Sparse-Cheetah-Fwd-Back | 2 | 2 | 6 |
| Walker2d-Rand-Params | 20 | 5 | N/A |

## D.4 IMPLEMENTATION

All experiments are carried out on 64-bit CentOS 7.2 with Tesla P40 GPUs. Code is implemented and run with PyTorch 1.2.0. One can refer to the source code in the supplementary material for a complete list of dependencies of the running environment.

Table 6: Hyperparameters used for training to produce Figure 4(a). Meta-batch size refers to the number of distinct tasks for computing the DML or contrastive loss at a time. Larger meta-batch size leads to faster convergence but requires greater computational power. For Fwd-Back environments, a meta-batch size of 4 suffices for stability and efficiency.

| Hyperparameters | Point-Robot | Mujoco |
|---|---|---|
| reward scale | 100 | 5 |
| discount factor | 0.9 | 0.99 |
| maximum episode length | 20 | 200 |
| target divergence | N/A | 0.05 |
| behavior regularization strength($\alpha$) | 0 | 500 |
| latent space dimension | 5 | 20 |
| meta-batch size | 16 | 16* |
| dml_lr($\alpha_1$) | 1e-3 | 3e-3 |
| actor_lr($\alpha_2$) | 1e-3 | 3e-3 |
| critic_lr($\alpha_3$) | 1e-3 | 3e-3 |
| DML loss weight($\beta$) | 1 | 1 |
| contrastive T | 0.5 | 0.5 |
| contrastive m | 0.9 | 0.9 |
| buffer size (per task) | 1e4 | 1e4 |
| batch size (sac) | 256 | 256 |
| batch size (context encoder) | 512 | 512 |
| g_lr(f-divergence discriminator) | 1e-4 | 1e-4 |
| transformer hidden size (context encoder) | 128 | 128 |
| multihead (if enabled) | 8 | 8 |
| reduction (batch attention) | 16 | 16 |
| transformer blocks (context encoder) | 3 | 3 |
| dropout (context encoder) | 0.1 | 0.1 |
| network width (others) | 256 | 256 |
| network depth (others) | 3 | 3 |

