# OpenReview forum: "Provably Improved Context-Based Offline Meta-RL with Attention and Contrastive Learning"
_ICLR.cc/2022/Conference — ICLR 2022 Submitted_

### Official Review · Reviewer_oBW4 · 2021-10-26

**Correctness:** 2
**Technical Novelty And Significance:** 3
**Empirical Novelty And Significance:** 1
**Recommendation:** 6
**Confidence:** 2

**Main Review:**

This paper provides a descent improvement for the SOTA offline meta RL algorithm, FOCAL. The main contribution is a combination of three existing techniques that are widely used in other domains, and therefore the novelty / significance is somewhat limited. The experiment results do show the improved test returns and reduced variance in all the six tasks, and the ablation studies are helpful to understand the benefit of each component and its advantage under distribution shift. Therefore, this work is marginally above the acceptance threshold.

The theoretical analysis is somewhat interesting, but I am not sure how much value it adds to the paper and the proof in Theorem 3.3 is not solid.

First, the equivalence between the matrix-form momentum contrast objective and the mean classifier loss is quite straightforward by observing Eq 8 as the log-soft-max loss with the logits being the linear product of latent vectors. The main theorem 3.2 in Section 3.3 is an application of Lemma 4.3, but I don’t find any details about how it is applied. While we can say L_{sup}^\mu (13) is closer to the supervised contrastive loss than the unsupervised contrastive loss (14), it is hard to argue that the former serves as a better surrogate simply because of that, neither (13) will learn a better task embedding than (14).

Second, I’m not sure if theorem 3.3 is rigorously proven in the appendix. A key step of the proof is Eq 31, but the argument in the text between 30 and 31 is rather vague. I do not see how 31 is derived. Even if Theorem 3.3 is true, it only shows that when the context encoder with a batch-wise gated attention is optimised, the variance of the embedding after the attention is smaller than the variance of equally weighted embedding before that layer. But it does not show whether it is smaller than the variance of the learned embedding if one optimises the original context encoder without the attention.

In the experiment section, it seems the contribution of the contrastive loss is only marginal from Table 1 and 2. How would FOCAL++ perform if we only use the batch-wise and seq-wise attention? Also, I suppose adding the two attention architecture increase the overall size of the context encoder. Will we improve the performance of FOCAL with a similar gain by using a bigger network with the original architecture?


**Summary Of The Paper:**

This paper proposes to improve a context-based offline meta-RL algorithm, FOCAL, by several modifications to the context encoder’s architecture and training objective. Specifically, it adds batch-wise gated attention and sequence-wise self-attention in the encoder network, and proposes to replace the classical unsupervised contrastive loss with the InfoNCE objective. Theoretical analysis is done to show the proposed objective is closer to the supervised contrastive loss and the variance of task embedding with an optimised encoder is smaller than the one without batch-wise gated attention. Experiments on a few common meta RL benchmarks and sparsified environments show the superiority of the method over FOCAL and a few other baselines.

**Summary Of The Review:**

The architecture improvement on FOCAL is a good contribution in practice, while the theoretical analysis part requires some work for more solid discussion and proof.

---

> ### Author Response · Authors · 2021-11-23
> **Response to Reviewer oBW4**
>
> We thank the reviewer for appreciating our well-executed experiments to show improved test returns and reduced variance, as well as ablations studies to help better understand the advantage of our method  under distribution shift and sparse reward.
>
> To address your questions:
>
> **Q1**: "First, the equivalence between the matrix-form momentum contrast objective and the mean classifier loss is quite straightforward by observing Eq 8 as the log-soft-max loss with the logits being the linear product of latent vectors. The main theorem 3.2 in Section 3.3 is an application of Lemma 4.3, but I don’t find any details about how it is applied. While we can say L_{sup}^\mu (13) is closer to the supervised contrastive loss than the unsupervised contrastive loss (14), it is hard to argue that the former serves as a better surrogate simply because of that, neither (13) will learn a better task embedding than (14).":
>
> **A1**: The RHS of Theorem 3.2 is derived from Lemma 4.3 by taking $\tau\rightarrow 0$ as we have no class collision when given full information of training task labels. The LHS of Theorem 3.2 holds naturally since by definition in Eqn 10, the supervised contrastive loss is the infimum of Eqn 9 for all possible $W$. Please check our General Response 3 for explanation of why (13) serves as a better surrogate to be optimized than (14).
>
> **Q2**: "Second, I’m not sure if theorem 3.3 is rigorously proven in the appendix. A key step of the proof is Eq 31, but the argument in the text between 30 and 31 is rather vague. I do not see how 31 is derived. Even if Theorem 3.3 is true, it only shows that when the context encoder with a batch-wise gated attention is optimised, the variance of the embedding after the attention is smaller than the variance of equally weighted embedding before that layer. But it does not show whether it is smaller than the variance of the learned embedding if one optimises the original context encoder without the attention.":
>
> **A2**: The proof of theorem 3.3 in the Appendix is simplified by assuming constant weight on the non-sparse set as well as the absolutely sparse set. Eqn 31 holds because to minimize the loss in Eqn 30, the encoder tends to push away embedded vectors of different tasks while maintain intra-task similarity, which means the learned expected cosine similarity should satisfy $x_{i'}\cdot\mu_{i} < x_{i}\cdot\mu_{i}, \forall i'\neq i$, hence the RHS of Eqn 31. However, since the absolutly sparse samples are shared across tasks by Definition 3.5 and 3.6, $(x_{i'} - x_{i})\cdot\mu_{i}$ is always 0, the batch-wise attention tends to devalue such samples to minimize the total objective, and therefore we have the LHS of Eqn 31.
>
> Without batch-wise attention module, all embeddings are equally weighted, hence we think our simplified proof in Appendix B.2 does provide insightful explanation on our observation in Table 2 and 6.
>
> **Q3**: "In the experiment section, it seems the contribution of the contrastive loss is only marginal from Table 1 and 2. How would FOCAL++ perform if we only use the batch-wise and seq-wise attention? Also, I suppose adding the two attention architecture increase the overall size of the context encoder. Will we improve the performance of FOCAL with a similar gain by using a bigger network with the original architecture?":
>
> **A3**: Thanks for the suggestion. We increased the encoder width of FOCAL to make its # of parameters larger than FOCAL++ and find that FOCAL++ still consistently outperforms FOCAL. Please check our General Response 2.
>
> Overall, we are really grateful for your comments and advice, which are all valuable in helping us improve the work. Hope our explanation and updated material address your concerns. Please let us know if you have any further comments or concerns or ways in which we can further improve our paper.

---

> > ### Comment · Reviewer_oBW4 · 2021-11-29
> > **Does not resolve my concerns in the theoretical part**
> >
> > Thanks for the authors' response with additional experiments. The additional comparison with a larger baseline model is reassuring that the architecture modification does bring performance improvement. The contribution of contrastive loss still seems marginal to me.
> >
> > Regarding the theoretical part, the authors' response does not provide more clarification to the original text in the paper.
> >
> > Being closer to the idea loss means a tighter upper bound but not necessarily a better surrogate function. One chooses to optimize a surrogate loss at the price instead of the original loss usually because some good properties of the surrogate loss, e.g. ease of optimization, better regularization. It is a bit vague to argument L_{sup}^\mu is better than L_{um} only because it's a tighter bound.
> >
> > For Theorem 3, it is not *proved* in the appendix even if it is "simplified". Statements involving approximation, assumption, and words like "tends to" should give quantified results and conditions. I would say the "proof" in B.2 is more like a thinking process rather than a proof of a theorem.

---

> > > ### Author Response · Authors · 2021-11-30
> > > **Reply**
> > >
> > > On the significance of Theorem 3.3:
> > > The result of Theorem 3.3 is demonstrated empirically in Figure 6, where the variance of FOCAL++ become smaller than FOCAL as sparse ratio exceeds a threshold.
> > >
> > >
> > > Q:Being closer to the idea loss means a tighter upper bound but not necessarily a better surrogate function. One chooses to optimize a surrogate loss at the price instead of the original loss usually because some good properties of the surrogate loss, e.g. ease of optimization, better regularization. It is a bit vague to argument L_{sup}^\mu is better than L_{um} only because it's a tighter bound.
> > >
> > > A: The reason we use the unsupervised contrastive loss rather than the supervised one is clarified between eqn 9 and eqn 10 in the rebuttal version. **In short, the supervised loss we try to bound cannot be computed directly in the Meta-RL setting since it requires knowing the complete set of task labels, including both training and testing tasks. Our construction of contrastive objective enables self-supervised task representation learning for task inference, without requiring access to full labels of all possible tasks, which is flexible and has better potential for generalization.** We agree that a tighter surrogate does not translate to better performance. But in experiments we show that our contrastive loss performs favorably compared to the traditional contrastive loss. Theorem 3.2 can be part of the reason. To fully explain the effectness of our loss requires further work, which is not the main focus of our paper.

---

### Official Review · Reviewer_UknQ · 2021-11-01

**Correctness:** 4
**Technical Novelty And Significance:** 2
**Empirical Novelty And Significance:** 3
**Recommendation:** 5
**Confidence:** 4

**Main Review:**

This paper makes a valuable and novel contribution in the sense that offline meta-RL is somewhat under-studied and needs better methods. I also appreciated the focused study on context-based methods, and it seems like the proposed modifications could be widely applicable. However, I have some concerns on experimental and theoretical results.

**Experiments**

The experimental design leave much to be desired.

1. I do not understand the logic behind "using sparse-reward data only to learn context encoders": I assume the policy is learned with dense offline rewards, which seems unrealistic — either the observed rewards are dense or they are sparse. This seems to stem from assuming that encoder and policy are decoupled, but this is likely false: the policy is trained given a particular encoder, and replacing the encoder with a different one will degrade performance. In any case, I wished the paper included results on the standard non-sparse environments.
2. Similarly, I missed common baselines in offline meta-RL, e.g., MACAW (Mitchell et al., 2021) or BOReL (Dorfman et al., 2020). This is an issue because it's unclear how strong the included baselines (not specifically designed for offline RL) are against those methods, and thus limits potential interest from the community. If instead FOCAL++ was competitive with or outperformed those methods on (sparse & dense reward) tasks, this would be a strong selling point for the paper.
3. It seems like the proposed components (Attention & MoCo) could be combined with other context-based offline meta-RL methods (e.g., offline RL^2). Is it the case? If yes, showing such results would significantly strengthen the submission.
4. On the positive side, I appreciated the discussion on MDP ambiguity and sparse rewards (p. 9), which emphasize the importance of Attention (and MoCo) when task identification is challenging.

**Theory**

1. I find the results of theorems 3.1 and 3.2 misguiding. Th. 3.1 is just a matter of renaming the i-th column of $W^\mu$ as $\mathbb{E}_k[z^k_i]$, and I don't see a contribution there. As I understand it, Th. 3.2 simply invokes the results of Saunchi et al., 2019, so it seems the contribution is also very minor.
2. Th. 3.3 is more interesting, especially for context-based methods in sparse-reward environments.

**Minor details**:

- Recent work has already used attention for meta-RL (c.f., claim in 1st bullet point of p. 3): for example, Wang et al., 2021 use a transformer-based policy to meta-learn on tasks in their proposed benchmark.
- On p. 3, before Eq. 1: "which usually can be factorized" is missing "which"?

**References**

- Mitchell et al., "Offline Meta-Reinforcement Learning with Advantage Weighting", ICML 2021.
- Dorfman et al., "Offline Meta Learning of Exploration", ArXiv 2020.
- Wang et al., "Alchemy: A benchmark and analysis toolkit for meta-reinforcement learning agents", ArXiv 2021.

**Summary Of The Paper:**

This submission tackles the problem of offline meta-RL, where the goal is to learn a policy that can quickly solve new tasks. This policy must be learned on an offline set of (pre-collected) trajectories from multiple tasks, unlike in online meta-RL where data collection and learning are interleaved.

Specifically, the authors focus on improving a previous method, FOCAL, in 2 aspects:

1. They add an attention mechanism to improve the estimation of the context vector identifying the task. This mechanism attends to both the transitions from one state (and rewards, actions) to the next within a trajectory, and across trajectories within a mini-batch from a single task. The authors show that this mechanism can reduce variance when estimating task context.
2. They propose to use MoCo (He et al., 2020) to improve and accelerate learning of the task encoder. They include theoretical results showing that MoCo better approximates a supervised objective.

They conclude with experimental results on point-mass and simple Mujoco tasks, where FOCAL++ compares favorably against FOCAL and contextual variants of meta-RL algorithms.

**Summary Of The Review:**

Overall, I think this paper proposes reasonable modifications for context-based offline meta-RL methods. But, I worry that their results focus too narrowly on FOCAL to be of wide interest to the ICLR community.

---

> ### Author Response · Authors · 2021-11-23
> **Response to Reviewer UknQ**
>
> We thank the reviewer for recognition of several key elements of our paper:
>
> 1. We pushes the limit of the current SOTA methods of context-based offline meta-RL problem, which is an important but relatively under-studied area.
> 2. Our proposed method (attention + contrastive learning for task inference) can be widely applicable
> 3. We shed light on the superiority of our proposed method in addressing MDP ambiguity and sparse reward through discussion in page 9.
> 3. Our proposed method is firmly grounded with mathematical motivation and proofs (e.g. Th. 3.3 to show our method achieves smaller variance of task representation in sparse-reward environments)
>
> To address your questions:
>
> - Experiments
>
> **Q1**: "I do not understand the logic behind "using sparse-reward data only to learn context encoders": I assume the policy is learned with dense offline rewards, which seems unrealistic — either the observed rewards are dense or they are sparse. This seems to stem from assuming that encoder and policy are decoupled, but this is likely false: the policy is trained given a particular encoder, and replacing the encoder with a different one will degrade performance. In any case, I wished the paper included results on the standard non-sparse environments":
>
>
> **A1**: Indeed our implementation of using sparse-reward data only to learn context encoders follows the key finding from FOCAL that on offline multi-task dataset, the gradients of task representation learning and Bellman updates can be decoupled. However, this doesn't mean the task representation and policy are fully decoupled -- the learned policy is still conditioned on the latent context. Nevertheless, we also added experiments on standard non-sparse environments, please check our General Response 2.
>
> **Q2**: "Similarly, I missed common baselines in offline meta-RL, e.g., MACAW (Mitchell et al., 2021) or BOReL (Dorfman et al., 2020). This is an issue because it's unclear how strong the included baselines (not specifically designed for offline RL) are against those methods, and thus limits potential interest from the community. If instead FOCAL++ was competitive with or outperformed those methods on (sparse & dense reward) tasks, this would be a strong selling point for the paper":
>
>
> **A2**: Please check our General Response 2 for comparision of FOCAL++ with MACAW and BOReL on standard non-sparse datasets.
>
>
> **Q3**: "It seems like the proposed components (Attention & MoCo) could be combined with other context-based offline meta-RL methods (e.g., offline RL^2). Is it the case? If yes, showing such results would significantly strengthen the submission":
>
> **A3**: We totally agree that our proposed components can be widely applicable to other COMRL methods, since it only involves learning robust task representations, which can be decoupled from learning of policy according to FOCAL. One could naively subsitute SAC or BRAC in FOCAL/FOCAL++ with other similar offline RL algorithms such as CQL [1]. We didn't include such experiments since we think this will not make much difference and has nothing to do with our core contribution as stated in our General Response 1.
>
> Offline RL^2 as suggested by Reviewer UknQ, however, is hard to realize since the original RL^2 is based on on-policy algorithms like PPO/TRPO which cannot be naively extended to offline scenario. Should the paper be accepted, we will consider adding such abalation studies if any offline algorithms that are fundamentally different from SAC/BRAC were spotted.
>
>
> - Theory
>
> **Q1**: "I find the results of theorems 3.1 and 3.2 misguiding. Th. 3.1 is just a matter of renaming the i-th column of W^{\mu} as \mathbb{E}_k[z_i^k], and I don't see a contribution there. As I understand it, Th. 3.2 simply invokes the results of Saunchi et al., 2019, so it seems the contribution is also very minor."
>
> **A1**: We have stressed and clarified our key contribution, as well as clarified our theoretical analyses in General Response. Please check our General Response 1 & 3.
>
> - Minor details
>
> **Q1**: "Recent work has already used attention for meta-RL (c.f., claim in 1st bullet point of p. 3): for example, Wang et al., 2021 use a transformer-based policy to meta-learn on tasks in their proposed benchmark."
>
> **A1**: Thanks for pointing us to this recent paper, we have made changes in our discussion of related work for "Attention in RL" and contribution (we missed it because these parts were drafted earlier than Neurips 2021). However, our work is still the first to consider attention for **offline** meta-RL.
>
> Overall, we are really grateful for your comments and advice, which are all valuable in helping us improve the work. Hope our explanation and updated material address your concerns. Please let us know if you have any further comments or concerns or ways in which we can further improve our paper.
>
> [1] Kumar, Aviral, et al. "Conservative Q-Learning for Offline Reinforcement Learning." arXiv preprint arXiv:2006.04779 (2020).

---

> > ### Comment · Reviewer_UknQ · 2021-11-28
> > **Reply**
> >
> > I thank the authors for their response, and will keep my original rating. My decision is mostly motivated by the limited contributions of Th. 3.1 and 3.2, and the lack of baselines in the original version of the paper. The updated results (from General Response 2) look promising, but I think they require more thorough discussion to justify acceptance (eg, can we compare convergence curves between methods? how do each method adapt when given access to a few online interactions?).

---

> > > ### Author Response · Authors · 2021-11-29
> > > **Reply**
> > >
> > > Hi, thanks for the quick feedback. For the convergence curves of FOCAL++ vs. MACAW&BOReL, please see https://drive.google.com/file/d/1bXWc-BmYHWvL1HUcHXFLGgG17pUxy_y4/view?usp=sharing as a result on the 6 MuJoCo environments. It's evident that FOCAL++ significantly outperforms MACAW and BOReL in terms of asymptotic performance while achieving a comparable or better convergence speed.
> > >
> > > For your comments about "require more thorough discussion" such as "how do each method adapt when given access to a few online interactions", we will see if some quick experiments or analyses can be done given the time constraint to provide more insight. However, we need to stress that the FOCAL/FOCAL++/MACAW are fundamentally fully-offline methods, which means the algorithms adapt at test time with offline data only, so online adapataion is not really the central focus of our paper, nor does it affect our key contribution of robust offline task representation learning for meta-RL.

---

> > > ### Author Response · Authors · 2021-11-30
> > > **Further Comments to Your New Questions**
> > >
> > > Convergence curves:
> > >
> > > On relatively easy tasks, such as Point-Robot-Wind, all three methods perform similarly. However, on other tasks, BOReL fails to learn. MACAW, on the other hand, achieves similar sample efficiency on Sparse-Point-Robot and Point-Robot-Wind, and even better sample efficiency on Half-Cheetah-Vel. However, MACAW perform inferior in terms of asymptotic performance on other tasks.
> > >
> > > Online adaptation:
> > > We want to stress that our fully offline setting is more relevant in reality, as direct online adaptation can be risky and expensive. During adaptation, the weight of all networks stay fixed, so models are not trained on collected data. This data is used only to generate the context, on which the policy and value is conditioned. And the only difference between online and offline adapatation is the quality of data. We can see the effect of adaptation data's quality on testing performance from Table 3 and FOCAL++ is rather robust to it even when using random data. Moreover, the data collected online by trained FOCAL++ most likely will not be random.

---

### Official Review · Reviewer_uEZk · 2021-11-02

**Correctness:** 4
**Technical Novelty And Significance:** 2
**Empirical Novelty And Significance:** 3
**Recommendation:** 6
**Confidence:** 3

**Main Review:**

Strengths:

1. The paper focuses on a very important problem in reinforcement learning: meta-learning. The ability to adapt to new tasks and
transfer learning is extremely important component of intelligence.

2. The authors firmly ground their approach with mathematical motivation and proofs.

3. The quantitative results on the given Mujoco dataset are competitive and show promise.

Weaknesses:

1. The conceptual novelty is limited. The paper effectivly just takes FOCAL and adds an attention mechanism and a contrastive loss.

2. The approach was evaluated on only one dataset. It is difficult to draw conclusions on the generality of the approach. An evaluation
on another meta-learning environment (such as perhaps Meta-World) would strengthen the case of the paper.

**Summary Of The Paper:**

In this paper, the authors focus on the problem of meta-learning for offline reinforcement learning. To this end, they build
on the FOCAL algorithm by adding to key contributions. First, they add an intra-task attention mechanism. Second, they add
inter-task contrastive learning. The authors provide a theoretical foundation for their approach and evaluate on control tasks
in MuJoco. They demonstrate competitive performance over the state of the art.

**Summary Of The Review:**

In general, the authors have proposed a modification to an existing algorithm, FOCAL, which incorporates attention and contrastive learning.
They evaluate on only one dataset. In spite of these weaknesses, the approach has strong mathematical foundations, and results on the Mujoco dataset look promising.

---

> ### Author Response · Authors · 2021-11-23
> **Response to Reviewer uEZk**
>
> We thank the reviewer for recognition of several key elements of our paper:
>
> 1. We focus on improving the generalization and transfer learning ability of the current offline meta-RL methods, which is extremely important component of intelligence
> 2. Our proposed method is firmly grounded with mathematical motivation and proofs
> 3. Our proposed method is supported by promising experimental result and quantitaive analyses on common meta-RL benchmarks.
>
> To address your questions:
>
> **Q1**: "The conceptual novelty is limited. The paper effectivly just takes FOCAL and adds an attention mechanism and a contrastive loss":
>
> **A1**: We have stressed and clarified our key contribution. Please check our General Response 1.
>
> **Q2**: "The approach was evaluated on only one dataset. It is difficult to draw conclusions on the generality of the approach. An evaluation on another meta-learning environment (such as perhaps Meta-World) would strengthen the case of the paper":
>
> **A2**: We thoroughly tested FOCAL++ on 6 MuJoCo datasets with sparse rewards, all of which are common meta-RL benchmarks and aligned with well-known papers like PEARL [1] and MQL [2], with one additional relabeled dataset for discussion of MDP ambiguity (Figure 5). As some reviewers suggested, we also added more experiments with other COMRL baselines on non-sparse environments. Please check our General Response 2.
>
> Overall, we are really grateful for your comments and advice, which are all valuable in helping us improve the work. Hope our explanation and updated material address your concerns. Please let us know if you have any further comments or concerns or ways in which we can further improve our paper.
>
> [1] Rakelly, Kate, et al. "Efficient off-policy meta-reinforcement learning via probabilistic context variables." International conference on machine learning. PMLR, 2019.
>
> [2] Fakoor, Rasool, et al. "Meta-Q-Learning." International Conference on Learning Representations. 2019.

---

### Official Review · Reviewer_yZzz · 2021-11-02

**Correctness:** 3
**Technical Novelty And Significance:** 2
**Empirical Novelty And Significance:** 3
**Recommendation:** 5
**Confidence:** 3

**Main Review:**

Strengths:
- Description of the method is reasonably easy to follow
- Interesting experiments on distribution shift and MDP ambiguity

Weaknesses:
- The scope of the paper is rather narrow (specifically context-based offline meta-RL, mostly focused on improving a single algorithm), which will probably limit impact; consider adding comparisons with [1] or [3]?
- The technical novelty is limited (essentially, adding attention)
- It's very difficult to interpret the significance of the main theoretical claim; can the authors justify why L_{sup} is a meaningful lower bound? Where does this loss come from? Unless I'm missing something, the solution of W = 0 would give zero loss, depending on the choice of \ell
- The writing gets very difficult to follow in the theoretical sections; variables are not always defined unambiguously and some theoretical statements lack justification/explanation

Other comments:
- The term "MDP ambiguity" doesn't come from [2], but [1] (as far as I know)
- "The intuition behind sequence-wise attention is that the attentive context encoder should in principle better capture the correlation in (s, a, s′, r) sequence related to task-specific reward function R(s, a) and transition function P (s′ |s, a), compared to normal MLP layers employed by common context-based RL algorithms." MLP layers are already universal function approximators, so it's not clear to me why self-attention is needed to learn a better representation of the tuple. If the tuple. Self-attention seems useful if the sequence is too long to fit into a regular MLP or if the sequence length is variable, but neither of these are the case here, as far as I can tell.
- It seems to me that the matrix form of the contrastive loss is an implementation detail, rather than a novelty, but maybe I'm missing something...
- The notation in section 3.2 and 3.3 is a bit confusing, because the same symbol (e.g. bold z) means a matrix in 3.2, but a single vector in 3.3.
- Without knowing what \ell is in Eq 9, it's hard to know what the equation means. Currently, it's not clear that this "supervised contrastive loss" is a meaningful objective. Assuming that \ell(0) = 0, doesn't the zero matrix give zero loss?
- I don't really follow the interpretation of the key vectors as classifiers- wouldn't the query classify whether or not particular keys are of the same class?
- Definition 3.1: It's unintuitive to me to define the loss for a classifier as the difference between their outputs, as opposed to e.g. KL divergence or cross entropy. Why was this form chosen?
- Are the queries formed by simple averaging, or the weighted average from self-attention?
- "Since we assume no access to the entire task set" This is strange to me- we're using the task set during training, right?
- The writing in Sec 3.3 is difficult to follow. In particular, I didn't realize that we were trying to come up with a surrogate objective for the "supervised loss" until the middle of page 6. It's also not clear to me that the supervised loss in Eq 9 is a meaningful thing to bound.
- if the constant is assumed to be zero, I'm not sure definition 3.5 makes sense- doesn't this mean all transitions have reward zero?
- Section 3.4 says that a proof for Theorem 3.3 is given in the Appendix, but the Appendix lists an "informal proof", which is a bit strange

[1] Dorfman & Tamar. Offline Meta-Learning of Exploration. 2020.
[2] Li et al. Multi-task Batch Reinforcement Learning with Metric Learning. 2020.
[3] Mitchell et al. Offline Meta-Reinforcement Learning with Advantage Weighting. 2021.

**Summary Of The Paper:**

The paper intends to improve a recent offline meta-RL algorithm (FOCAL) through a modified objective and architecture. The authors propose the use of attention both for up-weighting more informative transition tuples in a batch of adaptation data, as well as within the transition tuples themselves, in order to improve task inference in sparse reward settings. In addition, the authors modify the objective used in FOCAL to form a tighter bound on a "supervised" contrastive loss than prior work, though it's not totally clear how meaningful this supervised form of the loss is.

**Summary Of The Review:**

I have mixed feelings on this paper. Overall, I think the technical novelty and impact of theoretical contributions is potentially limited, but some of the experiments in distribution shift/MDP ambiguity are interesting. Currently I'm not comfortable accepting the paper on account of dubious theoretical contributions and limited scope/technical novelty, but if the authors can convincingly argue why their bound in Theorem 3.1 is meaningful (motivate the supervised version of the loss), add additional baselines to make the paper less specific to just FOCAL/COMRL, and improve the writing in the theoretical sections, I am open to accepting it (but could be convinced otherwise).

---

> ### Author Response · Authors · 2021-11-23
> **Response to Reviewer yZzz - Part 1**
>
> We thank the reviewer for recognition of several key elements of our paper:
>
> 1. The contribution and description of the method are well-delivered.
> 2. Interesting experiments on distribution shift and MDP ambiguity are included and well-executed.
>
> To address your questions:
>
> **Q1**: The scope of the paper is rather narrow (specifically context-based offline meta-RL, mostly focused on improving a single algorithm), which will probably limit impact; consider adding comparisons with [1] or [3]?
>
> **A1**: We have clarified our key contribution and provide more experiments as suggested. Please check our General Response 1&2.
>
>
> **Q2**: It's very difficult to interpret the significance of the main theoretical claim; can the authors justify why L_{sup} is a meaningful lower bound? Where does this loss come from? Unless I'm missing something, the solution of W = 0 would give zero loss, depending on the choice of \ell
>
> **A2**: For the first two questions, please check our General Response 3.
>
> For the last question, for classification tasks, the classifier $g=WE$ typically requires a normalized activation function (e.g. softmax) to estimate the probability that the data point belongs to each class. Therefore $W=0$ simply means assigning equal probability to all classes, which is clearly not a locally/globally optimal solution.
>
> **Q3**: The writing gets very difficult to follow in the theoretical sections; variables are not always defined unambiguously and some theoretical statements lack justification/explanation
>
> **A3**: We have updated section 3.3 according to our clarification in General Response 3, please check the rebuttal revision. Please check the second thread in which we make detailed explanation to your questions in "other comments".
>
> Overall, we are really grateful for your comments and advice, which are all valuable in helping us improve the work. Hope our explanation and updated material address your concerns. Please let us know if you have any further comments or concerns or ways in which we can further improve our paper.

---

> > ### Author Response · Authors · 2021-11-23
> > **Response to Reviewer yZzz - Part 2**
> >
> > For your other comments:
> >
> > **Q1**: "The term "MDP ambiguity" doesn't come from [2], but [1]"
> >
> > **A1**: Thanks for the note. Considering the fact that [2] (Sep 2019) appeared much earlier than [1] (Aug 2020) and discussed a very similar problem without using the exact terminology, we added both citations in the latest manuscript.
> >
> > **Q2**: "MLP layers are already universal function approximators, so it's not clear to me why self-attention is needed to learn a better representation of the tuple."
> >
> > **A2**: The interpretation of sequence-wise attention is not grounded by theoretical analyses, but more of emipirical exercise. As with [3], we found such attention to achieve a performance gain on almost all of our datasets, compared to vanillar FOCAL with even larger encoder networks (see also our General Response 2):
> >
> > |Algorithm| Sparse-Point-Robot | Point-Robot-Wind | Cheetah-Dir | Ant-Dir | Cheetah-Vel | Walker-2D-Params |
> > | -- | -- | -- | -- | -- | -- | -- |
> > | FOCAL++(sequence-wise, # of params < 0.41M, from Table 1) | **12.64** | **-5.09** | 1293.40 | **573.26** | **-140.63** | **375.67** |
> > | FOCAL(# of params = 0.47M) | 11.79 | -5.51 | **1408.26** | 454.31 | -195.34 | 347.22 |
> >
> > **Q3**: "It seems to me that the matrix form of the contrastive loss is an implementation detail, rather than a novelty"
> >
> > **A3**: Yes you are right, this is not a novelty, but applying this loss for task representation learning on COMRL, as well as our theoretical analyses are novel.
> >
> > **Q4**: The notation in section 3.2 and 3.3 is a bit confusing, because the same symbol (e.g. bold z) means a matrix in 3.2, but a single vector in 3.3.
> >
> > **A4**: Sorry but we didn't use notation z in section 3.2. If you are refering to the Z in Figure 3, it is not bold and used to represent the latent space dimension. The bold z represents single vector throughout the paper.
> >
> > **Q5**: Without knowing what \ell is in Eq 9, it's hard to know what the equation means. Currently, it's not clear that this "supervised contrastive loss" is a meaningful objective. Assuming that \ell(0) = 0, doesn't the zero matrix give zero loss?
> >
> > **A5**: As with [4], $\ell$ can take form of standard hinge loss $\ell(v)=\max [ 0, 1+\max_i(-v_i) ] $ and logistic loss $\ell(v) = \log_2(1+\sum_i \exp(-v_i))$. We use latter for our proof of Theorem 3.1, which is clarified in the latest manuscript.
> >
> > **Q6**: I don't really follow the interpretation of the key vectors as classifiers...
> >
> > **A6**: Thanks for pointing this out. We have made changes in the latest manuscript accordingly.
> >
> > **Q7**: Definition 3.1: It's unintuitive to me to define the loss for a classifier as the difference between their outputs, as opposed to e.g. KL divergence or cross entropy. Why was this form chosen?
> >
> > **A7**: Since we do not have the full set of task labels in meta-RL training phase, this form of contrastive objective enables self-supervised task representation learning for task inference, with better potential for generalization, especially when facing ood testing tasks.
> >
> > **Q8**: Are the queries formed by simple averaging, or the weighted average from self-attention?
> >
> > **A8**: They are formed by weighted averaged from batch-wise gated attention and seq-wise self-attention.
> >
> > **Q9**: "Since we assume no access to the entire task set" This is strange to me- we're using the task set during training, right?
> >
> > **A9**: Similar to A7, for meta-RL, we need to consider the whole underlying task set involving not only training set, but also testing set, and they may be ood. We have clarified this point in the latest manuscript.
> >
> > **Q10**: The writing in Sec 3.3 is difficult to follow. In particular, I didn't realize that we were trying to come up with a surrogate objective for the "supervised loss" until the middle of page 6. It's also not clear to me that the supervised loss in Eq 9 is a meaningful thing to bound.
> >
> > **A10**: Please see our General Response 3 and latest manuscript for better clarification.
> >
> > **Q11**: if the constant is assumed to be zero, I'm not sure definition 3.5 makes sense- doesn't this mean all transitions have reward zero?
> >
> > **A11**: By Definition 3.5, we only zero reward for **absolutely sparse transitions**, which are, for example, points that lie outside of **all circles** in Figure 5(a). Clearly this doesn't mean all transitions have reward zero.
> >
> > **Q12**: Section 3.4 says that a proof for Theorem 3.3 is given in the Appendix, but the Appendix lists an "informal proof", which is a bit strange
> >
> > **A12**: We would like to consider Appendix B.2 as a simplified rather than informal proof, which is reflected in our latest manuscript.
> >
> > [1] Dorfman & Tamar. Offline Meta-Learning of Exploration. 2020. [2] Li et al. Multi-task Batch Reinforcement Learning with Metric Learning. 2020. [3] Roberta Raileanu, et al. Fast adaptation to new environments via policy-dynamics value functions. [4] Arora, Sanjeev, et al. "A theoretical analysis of contrastive unsupervised representation learning."

---

> > > ### Comment · Reviewer_yZzz · 2021-11-29
> > > **Thank you for the response; still feel theoretical sections are significantly lacking**
> > >
> > > I really appreciate the authors' work in their revisions and responses. It is helpful to see comparisons with additional baselines; while the results are not earth-shattering, they provide added support for the effectiveness of the proposed algorithm and ultimately strengthen the paper, I think. I also appreciate the added ablation with the larger MLP version of FOCAL++, which is non-obvious and interesting.
> > >
> > > However, similarly to other reviewers, I'm still concerned about the theoretical contributions of the paper. I'm just not confident the theoretical sections are ready yet, and the relatively minor revisions made by the authors do not inspire confidence that, if the paper were accepted, they would be suitably revised for publication.
> > >
> > > I appreciate the authors modifying the text of the paper, but my original point about the clarity of the theoretical contributions stands. Beyond the relatively minor revisions in the response pdf, I feel the authors need to more significantly revise the writing of Section 3, particularly Section 3.3; for example, more discussion of Saunshi et al. is needed in the related work/buildup to the theoretical section (this work received only passing mention in the original submission), as this seems to be critical background needed to understand the formulation used in Section 3.3. The theoretical contributions here lean heavily on the framework described by Saunshi et al., but currently, the paper does not make clear which parts are novel and which are drawn from prior work.
> > >
> > > Further, the loss function in Equation 9 is simply not correct (comparing with Saunshi et al.), as far as I can tell. The authors use the difference between the representation vectors in Equation 9, but instead, the loss function should be defined as $\ell(\\{g(x)\_k - g(x)\_{k'}\\}\_{k \neq k'})$ for true class $k$ (see the definition of $L_{sup}$ from Saunshi et al.), which gives a $C-1$ dimensional vector for $C$-way classification; this has a totally different interpretation from the equation presented in the paper. This omission, as well as the general lack of discussion of the framework presented in Saunshi et al., are significant impediments to (my own, at the very least) understanding.
> > >
> > > I feel the authors' responses and revisions have improved the empirical contributions of the work, but have only reinforced the concerns about the theoretical contributions, which were the primary issue with the original contribution. Thus, I will keep my score.

---

> > > > ### Author Response · Authors · 2021-12-01
> > > > **Reply**
> > > >
> > > > Thanks for the feedback. For our general response, please check our "Additional General Response" on top.
> > > >
> > > > For your specific comments regarding Eqn 9, we noticed there is a missing $i\neq i'$ subscript on the RHS. Other than that, Eqn 9 is equivalent to the definition of $L_{sup}$ in Saunshi et al, up to different sets of notations. The only difference is that in context-based meta-RL, the positive/negative sample pairs are formed by sampling context samples $c_i$ and $c_{i'}$ from different tasks $T_i$ and $T_{i'}$ respectively. In Saunshi et al, since they focus on supervised learning instead of meta/multi-task learning, they use $x$ as sample and $c$ as class label, whereas we chose $c$ as context sample and $i$ as task label. Hope this explaination makes things clear.

---

### Author Response · Authors · 2021-11-23
**General Reponse - Part 1**

We sincerely thank the reviewers for taking time and providing insightful comments. We summarized several common issues raised and try to address them in this general response:

**1. The novelty and scope is somewhat limited (attention + contrastive learning on FOCAL)**

First of all, as Reviewer UknQ commented, we are trying to push the boundary of the current SOTA on COMRL problem, which is extremely important while relatively understudied, and focused but in-depth study can be a plus. Secondly, we agree that the building blocks of FOCAL++, namely attention + contrastive learning + FOCAL, are not new. However, the combination is by no means trivial. In fact, we are the first to make such design choices, for a clear purpose of learning more robust task representation, which significantly improves the current SOTA on challenging problems like MDP ambiguity and sparse reward. We provide **both** theoretical insight and experimental evidence on how our design works, which all reviewers think are interesting and insightful.

**2. The experiments only compare FOCAL++ with very limited baselines like FOCAL and MBML, and only on sparse reward datasets.**

We appreciate the suggestion. We have performed experiments on **non-sparse** version of the 6 Mujoco datasets for FOCAL++ and 2 competitive baselines: MACAW [1] and BOReL [2], and here are the final testing returns

|Algorithm| Sparse-Point-Robot | Point-Robot-Wind | Cheetah-Dir | Ant-Dir | Cheetah-Vel | Walker-2D-Params |
| -- | -- | -- | -- | -- | -- | -- |
| FOCAL++ | **12.96** | -5.39 | **1507.59** | **728.97** | -120.58 | **401.02** |
| MACAW | 11.37 | **-5.13** | 956.19 | 376.33 | **-117.04** | 312.7 |
| BOReL | 5.57 | -5.34 | 107.23 | -129.76 | -299.59 | 163.19 |

As one can see, even on non-sparse environments, **FOCAL++ compares favorably to other competitive baselines**. Besides, the training of MACAW is much slower due to the gradient-based MAML. BOReL underperforms in our fully-offline setting, since it is designed for online adaptation at test time, which exihibits severe bootstrapping error when using our offline data for test-time adaptation.

In addition, on **sparse** environments, we compared FOCAL++ and FOCAL with increased encoder network size (network width: 200 -> 600, resulting in 0.47M (FOCAL, MLP) vs. 0.41M (FOCAL++, Transformer) in terms of # of parameter) in response to Reviewer oBW4's question that if the performance gain is due to increased number of parameters:

|Algorithm| Sparse-Point-Robot | Point-Robot-Wind | Cheetah-Dir | Ant-Dir | Cheetah-Vel | Walker-2D-Params |
| -- | -- | -- | -- | -- | -- | -- |
| FOCAL++(from Table 1) | **12.96** | **-5.39** | **1470.52** | **719.77** | **-137.31** | **391.02** |
| FOCAL(0.47M) | 11.79 | -5.51 | 1408.26 | 454.31 | -195.34 | 347.22 |

To conclude, FOCAL++ consistently outperforms FOCAL with more parameters, suggesting that the performance gain is indeed due to our proposed design rather than simply increasing the # of parameters in the encoder network.

**Including these results in the main paper currently exceeds the page limit, we will make space should the paper be accepted.**

**3. The theoretical contribution in Section 3.3 is unclear**

Here we clarify what we want to deliver in 3.3.

Definition 3.1 represents a typical loss function incurred by classifier $g$ on supervised learning tasks, where $g = WE$, in which $W$ represents a downstream classifier and $E$ is an encoder function to form latent representations. **This "supervised contrastive loss" is the real objective one wants to optimize when given downstream tasks. However, in offline meta-RL setting, by only given limited training set, we aim to train a robust encoder $E$ without knowing the optimal $W$ (which requires full knowledge and labels of the underlying task distribution, including testing tasks).** For this purpose, we need a "surrogate" of Definition 3.1, for which we propose to use the Mean Task Classifier in Definition 3.2, which induces the Average Supervised Contrastive Loss in Definition 3.3. We notice that such construction is equivalent to the matrix-form momentum contrast in Eqn 8, as proven by Theorem 3.1. Moreover, it is easy to implement and takes full advantage of the multi-task structural information in OMRL setting. Finally, we compare it to a classifical unsupervised contrastive loss with only positive and negative pairs, used by prior meta-RL methods, and prove in Theorem 3.2 using lemma by Saunshi et al that our proposed contrastive loss acts as a better "surrogate" for the real objective one tries to optimize in Definition 3.1.

Once we obtain similarity-preserving task representations by optimizing the objective in Def 3.3, Appendix C in FOCAL sheds light on how conditioned meta-RL policy can be effectively learned.

[1] Dorfman & Tamar. Offline Meta-Learning of Exploration. 2020.

[2] Mitchell et al. Offline Meta-Reinforcement Learning with Advantage Weighting. 2021.

---

> ### Author Response · Authors · 2021-11-29
> **General Response - Part 2**
>
> As reviewer UknQ suggested, we also provide the convergence curves for FOCAL++ vs. MACAW&BOReL here: https://drive.google.com/file/d/1bXWc-BmYHWvL1HUcHXFLGgG17pUxy_y4/view?usp=sharing, on the 6 MuJoCo environments.  It's evident that FOCAL++ significantly outperforms MACAW and BOReL in terms of asymptotic performance while achieving a comparable or better convergence speed.

---

### Author Response · Authors · 2021-11-23
**Updates**

Dear Readers,

We have updated our final rebuttal manuscript. To summarize the work done for our rebuttal:

- Manuscript (revised text in blue)

  - As a response to the concern raised by serveral reviewers that the theotical significance and contribution of the paper is unclear, we revised Section 3.3 to better showcase our thought process related to the definitions and theorems, and to reflect our clarification in General Response 3.

  - Minor changes addressing technical details such as the mis-interpreted query and key vectors, suggested by Reviewer yZzz; also missing citations and some grammatical errors suggested by Reviewer UknQ.

- Replies to the reviewers

  - We summarized 3 major and common issues raised by the reviewers and directly address them in the following General Response. We also replied to each reviewer regarding their own questions in a separate thread. In particular, we provided additional experimental evidence involving three competitive baselines, MACAW, BOReL and FOCAL with larger encoder networks on non-sparse and sparse Mujoco environments respectively, and **demonstrate the superiority of our proposed method**.

Again thank you all for your valuable input. We have done our best based on the previous reviews to work towards a stronger submission. Hope the rebuttal can help make things more clear. We look forward to your comments and final decision.

---

### Author Response · Authors · 2021-12-01
**Additional General Response**

We thank the reviewers' timely feedback after seeing our rebuttal revision. **The reviewers agree that the added experiments comparing FOCAL++ vs. other COMRL baselines have strengthened the paper, and significantly widen the scope to be of interest to the ICLR community.** However, reviewers such as yZzz and oBW4 still have concern about the theoretical contribution and soundness of our paper, especially Section 3.3 in the paper. We hereby make the following comments as a new general reponse:

We would like to consider our discussion in Section 3.3 **a bonus rather than a liability** of this paper. Throughout the paper and rebuttal, we made it clear that the key contribution of this paper is an improved framework of SOTA COMRL algorithm, which is grounded by meticulous empirical study as well as theoretical analyses. **We never claim that the theory in Section 3.3 is original**, as we borrowed tools from works like Saunshi et al to shed light on why our proposed method work better. And we do think the analyses work as intended: they help illustrate the thought process of how we came up with this improved framework and why it helps FOCAL++ outperform in terms of asymptotic performance (Table 1) and lower variance (Table 2). **As theoretical analyses are not a must for similar RL algorithm papers published at top conferences (e.g. [1] and [2])**, we would like to think Section 3.3 as a nice-to-have plus rather than weak spot of the paper.

Overall we are really grateful for all the comments received, and tried our best to address the reviewers' questions, no matter major or minor. We encourage the readers to focus on the bright side of our work, particularly the carefully designed empirical studies, with theoretical insight. We look forward to your final decision.

[1]: Rakelly, Kate, et al. "Efficient off-policy meta-reinforcement learning via probabilistic context variables." International conference on machine learning. PMLR, 2019.
[2]: Li, Lanqing, Rui Yang, and Dijun Luo. "FOCAL: Efficient Fully-Offline Meta-Reinforcement Learning via Distance Metric Learning and Behavior Regularization." International Conference on Learning Representations. 2020.

---

### Decision · Program_Chairs · 2022-01-20

**Decision:**

Reject

**Comment:**

The paper addresses the problem of offline meta reinforcement learning. The authors build on the FOCAL algorithm, adding intra-task attention and inter-task contrastive representation learning objectives. The resulting FOCAL++ algorithm outperforms several strong baseline, including FOCAL and a theoretical analysis attempting to show that FOCAL++ provably improves on FOCAL is included.

Reviewers agreed that the novelty of the proposed approach is limited since attention and contrastive representation learning have been used in the closely related (online) meta-RL setting. At the same time reviewers agreed that the results in the paper and the rebuttal show that FOCAL++ improves on a strong set of baselines.

The main shared concern was regarding the significance and validity of the theoretical analysis. After considering the rebuttal reviewers voting for both acceptance and rejection were in agreement that there are issues with the theoretical analysis/justification. While we agree with the authors that the algorithmic and experimental part of the paper is strong, we have to base our decision on the state of the whole paper. In the end we decided not to accept the paper because 1) the paper put a significant focus on a theoretical argument the reviewers found problematic and 2) the authors did not modify the paper to sufficiently address these concerns during the available window.